# In-depth analysis of obesity-associated changes in adipose tissue-derived mesenchymal stromal/stem cells and primary cilia function

Nina-Naomi Kreis ✉, Alexandra Friemel, Andreas Ritter, Anna Elisabeth Hentrich, Esther Siebelitz, Frank Louwen & Juping Yuan

Adipose tissue-derived mesenchymal stromal/stem cells (ASCs) possess regenerative potential. Obesity induces a pro-inflammatory environment that compromises their function. Here we investigate how obesity affects ASC biology, focusing on primary cilia. Our data show that obesity alters ASC gene expression, particularly in pathways related to the extracellular matrix, transforming growth factor-β (TGFβ) signaling, cell motility, and differentiation. The gene levels of regulatory factor X2 (*RFX2*) and adenylate cyclase 3 (*ADCY3*), important for ciliary biogenesis, are downregulated in obese ASCs. TGFβ treatment significantly decreases the expression of *RFX2* and *ADCY3* in lean ASCs. Knockdown of *ADCY3* reduces primary cilium length, whereas pharmacological activation restores it and improves cell motility. These results suggest that obesity impairs ASC ciliary function, contributing to defective adipogenesis and reduced regenerative capacity. Restoring ADCY3 activity partially rescues ciliary integrity and cellular function, highlighting the role of primary cilia in ASC dysfunction and offering potential therapeutic targets for obesity-related disorders.

Adipose tissue (AT) has emerged as one of the most promising sources of mesenchymal stem cells (MSCs), owing to their abundance, ease of accessibility, low immunogenicity, regenerative potential, and relatively minimal ethical concerns[1,2]. ASCs, AT-derived MSCs, offer great promise for regenerative medicine. They can modulate the immune response, stimulate angiogenesis, and differentiate into multiple cell lineages to maintain or repair damaged tissues and organs, with subcutaneous AT being the most practical and clinically relevant source of ASCs[2]. Accordingly, they have been extensively studied and partially approved for therapeutic applications targeting different diseases[3,4]. Latest advances have enabled innovative strategies to reprogram human somatic cells into chemically induced pluripotent stem cells (hCiPSCs)[5] enabling the first-in-human phase I clinical trial for the treatment of type 1 diabetes with autologous transplantation of hCiPSCs-islets derived from abdominal subcutaneous ASCs[6].

ASCs influence the local and systematic environment through immune modulation and tissue homeostasis, mediated by direct cell-cell interactions and the secretion of multiple bioactive factors[7,8]. They secrete a variety of soluble and vesicular factors that remodel the extracellular matrix (ECM), a process crucial for their own function and regulation[9–11]. The ECM, in turn,

interacts with cells through transmembrane receptors that relay external cues and activate intracellular signaling pathways involving the actin cytoskeleton[12]. ASCs detect environmental stimuli through surface receptors, notably via a specialized, immotile sensory organelle called the primary cilium[13–15]. The primary cilium mediates diverse extracellular signals, acting as a transduction platform for a multitude of pathways, regulating proliferation, migration, and differentiation, and maintaining tissue homeostasis, growth, and development[16–18]. Ciliary signaling relies on various surface receptors within its membrane, including G protein-coupled receptors (GPCRs), along with their downstream effectors and second messengers[13,19].

The dramatic rise in obesity prevalence presents a significant global challenge, as it is linked to various diseases, including cardiovascular diseases, type 2 diabetes, multiple types of cancer, and an overall higher mortality rate[20,21]. It is well established that obesity causes changes in ASC's multipotency, proliferation, differentiation, and migration capacity, as well as energy metabolism by reducing their functionalities[7,22], suggesting that the health status of the donor should be considered to ensure the safe use of ASCs. Epigenetic changes of ASCs from individuals with obesity[23] as well as

Department of Gynecology and Obstetrics, Obstetrics and Prenatal Medicine, University Hospital Frankfurt, J. W. Goethe-University, Frankfurt, Germany.
✉e-mail: kreis@em.uni-frankfurt.de

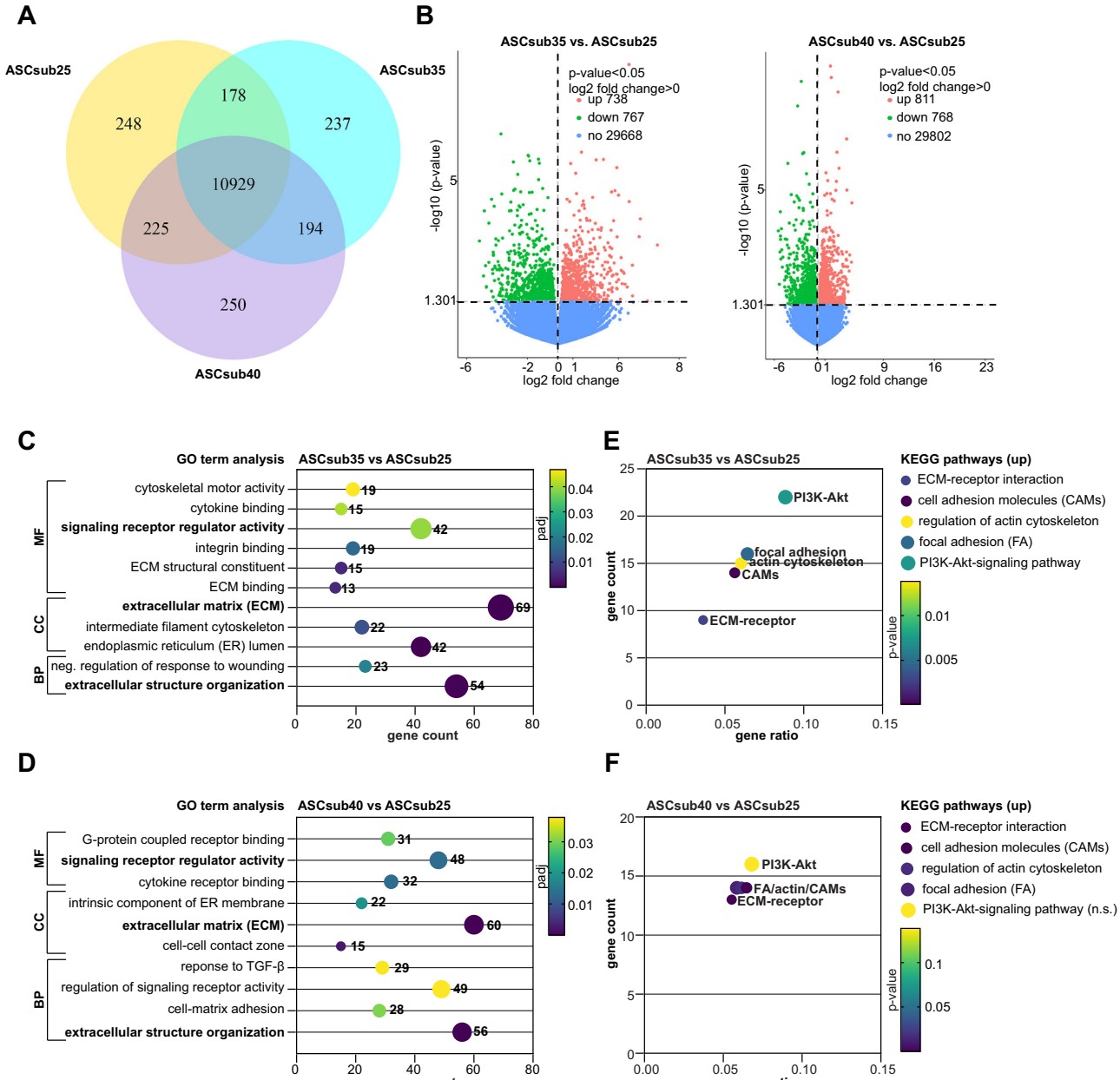

**Fig. 1 | Transcriptomic analysis of subcutaneous adipose tissue-derived mesenchymal stromal/stem cells (ASCsub).** Total RNAs were extracted from each ASC sample of individual donors with a body mass index (BMI) below 25 (ASCsub25, n = 5), equal or above 35 (ASCsub35, n = 5), and equal or above 40 (ASCsub40, n = 5) for transcriptomic analysis. **A** Venn diagram of co-expressed genes. **B** Volcano plots of significantly differentially expressed genes of ASCsub35 vs. ASCsub25 (left plot) and ASCsub40 vs. ASCsub25 (right plot). Upregulated genes are depicted in red, downregulated genes in green, and non-changed genes in blue. $p < 0.05$. Significantly enriched "Gene Ontology" (GO) pathways are presented for ASCsub35 compared to ASCsub25 (**C**) and for ASCsub40 compared to ASCsub25 (**D**), sub-clustered into three ontological domains: molecular function (MF), cellular component (CC), and biological process (BP). The adjusted $p$ value (padj) is displayed by a color code. Significantly enriched "Kyoto Encyclopedia of Genes and Genomes" (KEGG) pathways (except for PI3K-Akt ASCsub40 vs. ASCsub25) are presented for ASCsub35 compared to ASCsub25 (**E**) and for ASCsub40 compared to ASCsub25 (**F**). The $p$ value is indicated by a color code. Abbreviations: neg. negative, PI3K-Akt phosphoinositide 3-kinase - protein kinase B, TGFβ transforming growth factor-β.

an obesogenic epigenetic memory function of the AT were reported[24]. Moreover, we have studied ASCs derived from subcutaneous, visceral, and mammary AT from donors with obesity[14,25–28]. We have shown that obesity has adverse effects on subcutaneous and visceral ASCs, which are defective in ciliary biogenesis and Hedgehog signaling[14,15,26]. However, the molecular mechanisms by which ASCs are dysfunctional and their cilia are defective in obesity are not well investigated. Given that reactivation of ciliated cells and ciliary functions may offer a strategy to combat obesity and its related diseases[15], this study further characterizes ASCs from subcutaneous AT of donors with obesity. We performed whole-transcriptome analysis to identify deregulated genes and signaling pathways, with a focus on the functional integrity of primary cilia.

## Results

### Obesity leads to transcriptional reprogramming of ASCs

To investigate the effects of obesity on subcutaneous ASCs in more detail, total RNA was extracted from five individual ASC samples per group: lean (BMI < 25, ASCsub25), obese 35 (BMI ≥ 35, ASCsub35), and obese 40 (BMI ≥ 40, ASCsub40), and subjected to whole-genome RNA sequencing (RNA-seq). The Venn diagram (Fig. 1A) highlights overlapping and unique

genes of the three groups. The volcano plots (Fig. 1B) show significantly deregulated genes (*p* < 0.05). Compared to ASCsub25, ASCsub35 showed 738 significantly up- and 767 downregulated genes (Fig. 1B, left panel), whereas ASCsub40 demonstrated 811 significantly up- and 768 downregulated genes (Fig. 1B, right panel). Gene Ontology (GO) terms are categorized into three main ontological domains: biological process (BP), cellular component (CC), and molecular function (MF). Interestingly, our analysis revealed that obesity significantly altered the "signaling receptor regulator activity" of the MF, "ECM" of the CC, and "ECM structure organization" of the BP in both ASCsub35 and ASCsub40 populations (Fig. 1C, D). In the ASCsub35 group, four GO terms were directly associated with the ECM. To better illustrate the extent of gene overlap among these ECM-related GO terms (Fig. S1A), we used the web-based tool InteractiVenn[29]. The ASCsub40 group also enriched ECM-related GO terms, along with additional differentially expressed genes related to cell-matrix adhesion and response to transforming growth factor-β (TGFβ) (Fig. 1D and Fig. S1B). Moreover, the "Kyoto Encyclopedia of Genes and Genomes" (KEGG) pathway analysis highlighted the most significantly upregulated genes in both groups of obese ASCs, specifically, in ECM-receptor interaction, cell adhesion molecules (CAMs), regulation of the actin cytoskeleton, and focal adhesion (FA), compared to lean ASCs (Fig. 1E, F).

## Altered ECM genes in obese ASCs

Obesity-induced AT expansion requires ECM remodeling, which is associated with hypoxia and inflammation[30]. These conditions, in turn, upregulate ECM-related gene expression, particularly collagens, leading to its excessive accumulation and increased rigidity in obese AT[30]. ECM-related genes that are differentially expressed in both obese groups (ASCsub35 and ASCsub40), compared to ASCsub25, are depicted in Fig. S1C. Notably, they abundantly expressed various collagen genes (*COL*), including *COL11A1*, *COL12A1*, *COL4A6*, and *COL5A2* (Fig. S1D). In addition, the expression of crucial ECM regulator genes was enhanced (Fig. S1E), such as *ADAMTS12* (a disintegrin and metallopeptidase with thrombospondin type 1 motif 12), *HMCN1* (hemicentin 1), *MMP16* (matrix metallopeptidase 16), and *HAPLN1* (hyaluronan and proteoglycan link protein 1) (Fig. S1E). Importantly, *ANOS1* (anosmin-1), a TGFβ-regulated gene involved in endothelial cell migration and angiogenesis[31], *GAS6* (growth arrest-specific 6), which is related to cell adhesion, obesity, insulin resistance, and inflammation[32], and *WISP1* (WNT1-inducible signaling pathway protein 1), which is an adipokine associated with obesity, insulin resistance, and fibrosis, and a downstream target of TGFβ[33], were highly expressed in obese ASCs (Fig. S1F). The enhanced expression of *COL11A1*, *COL5A2*, and *GAS6* was corroborated with RT-PCR (Fig. S1G). These results highlight that ECM-related genes, particularly regulated by TGFβ, are deregulated in obese ASCs, which could lead to alterations of the ECM composition, structure, regulation, and function. This may, in turn, negatively affect the properties and functionalities of ASCs.

## Deregulated genes of the transforming growth factor β pathway in ASCs derived from donors with obesity

It is well established that TGFβ is increased during obesity[34,35]. This signaling pathway is crucial for regulating cell growth, differentiation, apoptosis, and migration[36]. To address if the TGFβ pathway is altered in our obese ASC cohorts, we further examined its related genes in RNA-seq data (Fig. 2A, B). Interestingly, three genes were enhanced expressed in both obese groups, *CDKN2B* (cyclin-dependent kinase inhibitor 2B, p15), *ITGB5* (integrin beta 5), and *THBS1* (thrombospondin 1) (Fig. 2A–C). Notably, ITGB5[37] and THBS1[38] are both able to activate the latent TGFβ1. In comparison with ASCsub25, ASCsub35 cells highly expressed several genes promoting the TGFβ pathway, such as *LPXN* (leupaxin), *LTBP1*, encoding the latent TGFβ binding protein 1, participating in TGFβ1 activation via integrins[39], and the TGFβ1-activating transcription factor gene *SOX9*, encoding SRY-box transcription factor 9 (Fig. 2D). In contrast, *TGFBR3* (TGFβ receptor 3,

betaglycan), the most abundantly expressed TGFβ-receptor lacking kinase activity, which enhances TGFβ signaling and mediates interactions with other proteins[40], however, was reduced in ASC35sub cells (Fig. 2E). Similarly, *TAB1* (TGFβ activated kinase 1 binding protein 1) and *CLEC3B* (C-type lectin domain family 3 member B, tetranectin) were also significantly downregulated in ASCsub35 (Fig. 2E). Looking more closely at the ASCsub40 group, we observed further altered genes (Fig. 2B). In particular, *ADAM17* (a disintegrin and metallopeptidase 17), which promotes epithelial to mesenchymal transition and fibrosis via TGFβ signaling[41], was increased. Additionally, *ASPN* (asporin), an important protein in the ECM linked to collagen fibrillogenesis[42], and *LTBP3* (latent TGFβ binding protein 3) that can bind fibrillin and TGFβ[39], also showed increased expression. Unexpectedly, *SMAD3* (mothers against decapentaplegic homolog 3), a prominent downstream target of TGFβ, was downregulated (Fig. 2F). These results strongly indicate that regulators and downstream effectors of TGFβ signaling are deregulated in obese ASCs. Since KEGG pathway analysis revealed an upregulation of actin cytoskeleton regulation and focal adhesion, and considering that TGFβ can influence cytoskeletal dynamics by regulating actin filament organization[43], we next focused on the altered genes within motility and migration pathways.

## Motility and migration genes are affected by obesity

ASCs have been reported to be able to migrate into injured areas, differentiate, and repair damaged tissue[2]. Taking into account that donor obesity may affect their therapeutic potential[7], we took a closer look at the differentially expressed genes related to motility and migration, which are significantly altered in both groups (ASCsub35 and ASCsub40) compared to ASCsub25 (Fig. 3A). Intriguingly, the genes *CDH1* (cadherin 1 or E-cadherin, an epithelial cell marker) and *CDH2* (N-cadherin, a mesenchymal cell marker) were increasingly expressed (Fig. 3B). Cadherins, transmembrane glycoproteins, play a major role in tissue homeostasis, cell-cell adhesion, and epithelial to mesenchymal transition, where E-cadherin expression is commonly lost[44]. Increased *CDH1* could result from hypoxia, which is a hallmark of AT in obesity[15], whereas enhanced *CDH2*, could be induced by various signaling pathways, including TGFβ[44]. *FN1* (fibronectin) and *LRP12* (low-density lipoprotein receptor related protein 12) were also significantly increased in ASCs derived from donors with obesity (Fig. 3C). Fibronectin is a major ECM glycoprotein involved in cell migration, tissue repair, and adipogenesis, and it interacts with several cell surface receptors, linked to integrin and focal adhesion dynamics[12,45]. Enhanced fibronectin may result in excessive accumulation and rigidity of ECM elements in obese AT[30]. *LRP12*, an endogenous transmembrane inhibitor for integrin activation, binds to the cytoplasmic tail of integrin α4 (ITG4A), and disturbs the association of talin (integrin-actin linker) with integrin β1 or β7, keeping integrin in its inactive state[46]. Besides *LRP12*, *ITG4A* is also enhanced in the transcriptomic data (Fig. 3A), both could contribute to the reduced migration capacity. The genes that were downregulated included *CXCL12* (CXC motif chemokine 12), *IFITM1* (interferon induced transmembrane protein 1), and *PGF* (placental growth factor) (Fig. 3D). Interestingly, the reduced expression of CXCL12 is contributed to impaired migration of ASCs[47]. Additional RT-PCR analyses were performed and validated an upregulation of *CDH1*, and downregulation of *CXCL12* as well as *PGF* in ASCs from donors with obesity (Fig. 3E). These data provide an explanation why obese ASCs are hampered in motility and migration capability[7,14].

## Deregulated genes of ciliary biogenesis in obese ASCs

The primary cilium is one of the most important mediators of extracellular signals and the cell has several pathways through which the length of the primary cilium can be modulated[48]. We have reported that primary cilia are crucial for various functionalities of ASCs, including their motility and migration, and that obesity shortens cilia rendering them dysfunctional[14,15]. To further underline that observation, we stained ASCs from ASCsub25, ASCsub35, and ASCsub40 for the ciliary markers acetylated tubulin (ace tubulin, green) and ARL13B (ADP-ribosylation factor-like GTPase 13B, red) (Fig. 4A), and measured the cilium length. The analysis showed that

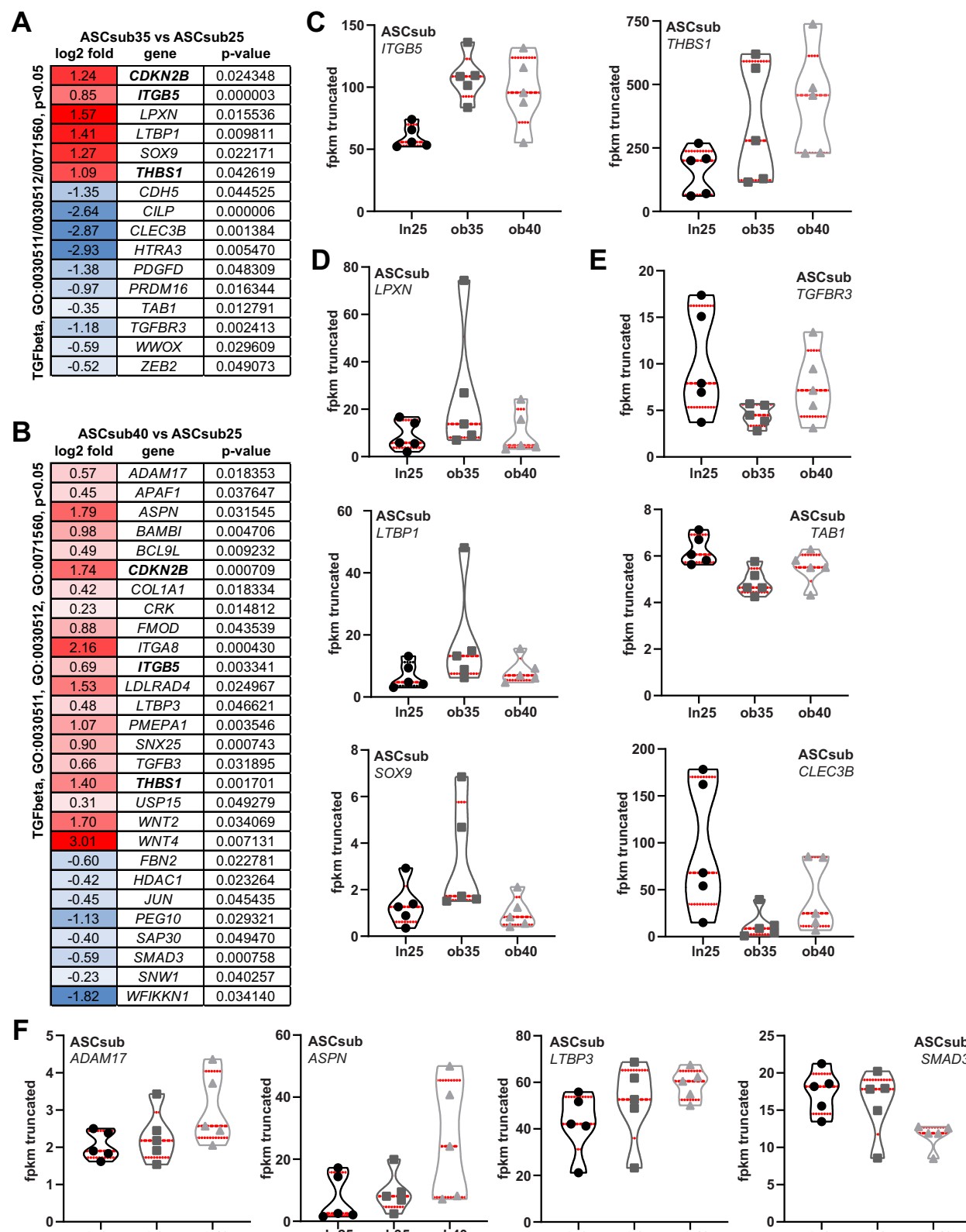

**Fig. 2 | Deregulated genes of the transforming growth factor-β pathway.** The heatmaps depict significantly ($p < 0.05$) differentially expressed transforming growth factor-β (TGFβ) genes with the gene ontology terms GO:0030511, 0030512, and 0071560 of ASCsub35 vs. ASCsub 25 (**A**) and of ASCsub40 vs. ASCsub 25 (**B**). Significantly enriched genes have a red color code and reduced genes are encoded in blue. Red and blue colored genes are sorted alphabetically. **C–F** Truncated violin plots present selected genes that are expressed differentially. Values reflect the fragments per kilobase per million mapped fragments (fpkm) of genes from the RNA-seq data. Each violin plot displays the median (central dashed line) and quartiles (upper and lower dotted lines).

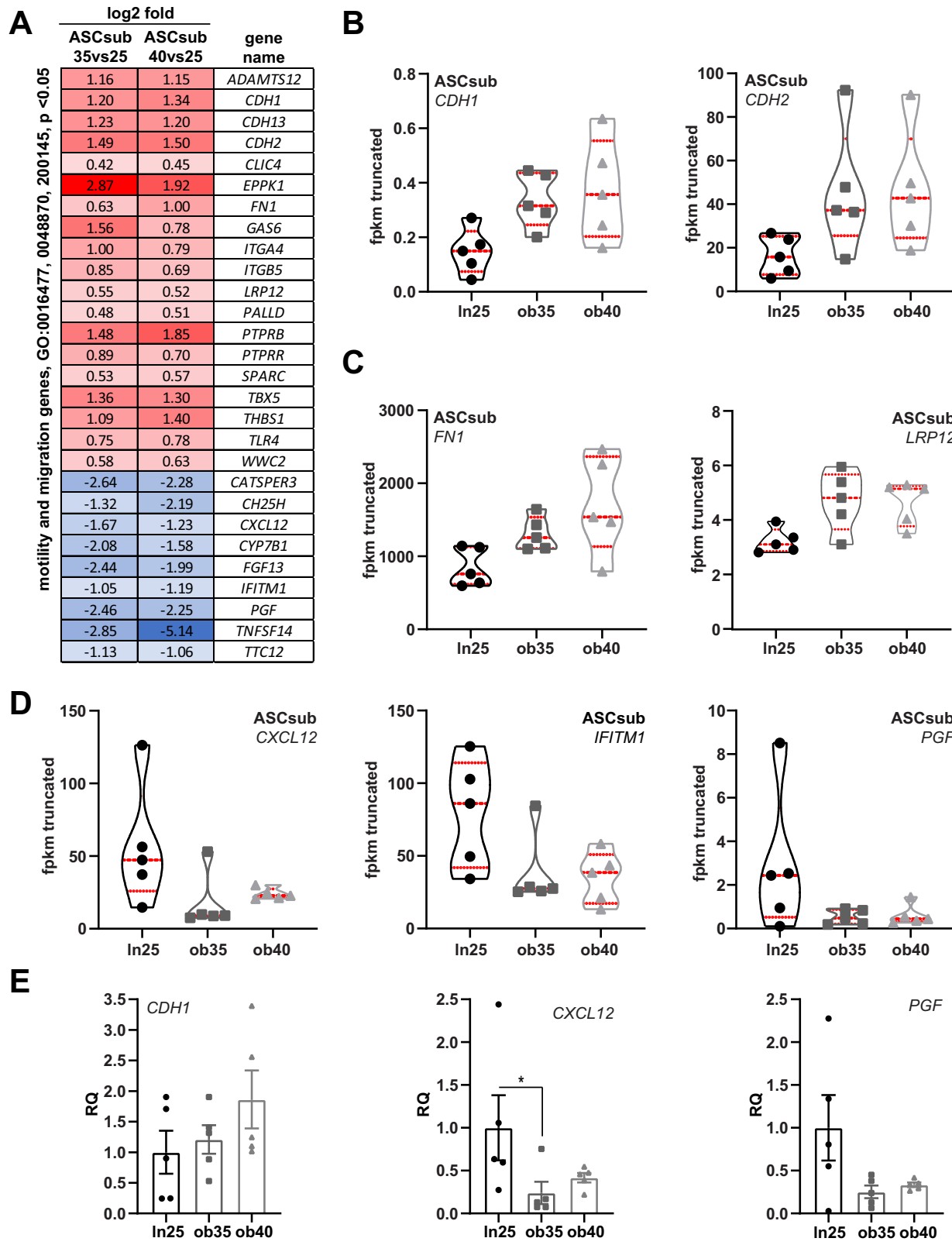

primary cilia of ASCsub35 as well as of ASCsub40 were shortened (Fig. 4B). We used CiliaCarta to identify known ciliary genes[49] within the transcriptomic data. Cilium-related genes were significantly altered in both groups (Fig. 4C), and even more in ASCsub35 (Fig. 4D). In particular, the gene expression of *GLI1* (glioma-associated oncogene homolog 1), which encodes an important regulator of the Hedgehog signaling pathway that is

exclusively transduced through the primary cilium[50], and *MYH10*, encoding the myosin heavy chain 10, required for centriole migration during cilium biogenesis[51], was enhanced (Fig. 4E), whereas the genes *PKIG* (protein kinase inhibitor gamma), a protein kinase A inhibitor leading to increased intracellular cAMP levels[52], and *SNX10*, encoding the sorting nexin 10, involved in the regulation of the vesicular trafficking and primary cilia

**Fig. 3 | Motility and migration genes are affected. A** The heatmap shows significantly ($p < 0.05$) differentially expressed motility and migration genes with the gene ontology terms GO:0016477, 0048870, and 200145 in both ASC subgroups of donors with obesity. Significantly enriched genes have a red color code and reduced genes are encoded in blue. Red and blue colored genes are sorted alphabetically. **B–D** Truncated violin plots present selected motility and migration genes that are differentially expressed. Values reflect the fragments per kilobase per million mapped fragments (fpkm) of genes from the RNA-seq data. Each violin plot displays the median (central dashed line) and quartiles (upper and lower dotted lines). **E** Relative gene levels of *CDH1* (cadherin 1), *CXCL12* (C-X-C motif chemokine ligand 12), and *PGF* (placental growth factor) are shown for lean and obese ASC subgroups (ln25, ob35 and ob40). The results were obtained from five individual samples in each group and presented as relative quantification (RQ) with SEM. *GAPDH* served as endogenous control. Ordinary one-way ANOVA followed by Dunnett's multiple comparisons test was used to assess statistical significance relative to ln25 (*CDH1*). For non-Gaussian data (*CXCL12*), the Kruskal-Wallis test followed by Dunn's post hoc test was applied. If variance homogeneity was not met (*PGF*), Welch's ANOVA with Dunnett's T3 multiple comparisons test was used. *$p < 0.05$.

membrane extension[53], were downregulated (Fig. 4F). Compared to ASC-sub25, ASCsub35 showed significant higher gene levels of *MYO7A* (myosin 7 A) and *HHIP* (Hedgehog interacting protein), genes linked to ciliopathies[54] and Hedgehog signaling[55] (Fig. 4G). Of particular interest, the gene levels of two crucial ciliary regulators, adenylate cyclase 3 (*ADCY3*), and clusterin associated protein 1 (*CLUAP1* or IFT38, intraflagellar transport protein 38) were reduced in obese ASCs (Fig. 4H). Further RT-PCR analysis underlined that *GLI1* was enhanced, whereas *SNX10*, *ADCY3*, and *CLUAP1* were decreased in obese ASCs (Fig. 4I).

### Reduced adipogenic differentiation of obese ASCs
Functional cilia are important for adipogenic differentiation of MSCs and ASCs[56,57] and as previously described by our group, shortened cilia of ASCs from obese donors were unable to respond dynamically to osteogenic differentiation stimuli[14]. To examine the capability of adipogenic differentiation, ASCs derived from lean donors (Fig. S2B, 1st and 2nd panels) or from donors with obesity (3rd and 4th panels) were incubated without or with adipogenic differentiation medium for 14 days for further analyses (Fig. S2A). Treated ASCs were stained with the lipid droplet dye BODIPY™, the cytoskeleton marker phalloidin, and the DNA dye DAPI for evaluating adipocytes. In comparison with control ASCs (Fig. S2B, 2nd panels), obese ASCs showed much less adipogenic differentiation capacity upon stimulation (Fig. S2B, 4th panels). These results were further underscored by brightfield images and Oil Red O staining (Fig. S2C, D). Evaluation revealed that only 16.99% of ASCs derived from donors with obesity were differentiated, compared to 56.48% of ASCs form lean donors (Fig. S2E). Moreover, total RNAs from these ASCs were extracted for gene analysis. Compared to lean ASCs, the expression of key adipogenic differentiation genes[58,59], including the early differentiation gene *PPARG* (peroxisome proliferator-activated receptor gamma), and its late effectors, *ADIPOQ* (adiponectin), *FABP4* (fatty acid-binding protein 4), and *LPL* (lipoprotein lipase), was significantly reduced (Fig. S2F). To meet the tri-lineage differentiation criteria defined by the International Society for Cellular Therapy (ISCT)[60], we also performed osteogenic and chondrogenic differentiation for 21 days. Both differentiation capacities were reduced in ASCs derived from donors with obesity, as indicated by decreased expression of the master transcription factors *RUNX2* (runt-related transcription factor 2) for osteogenesis and *SOX9* for chondrogenesis (Fig. S2G, H). These data underscore the observation that obesity impairs the differentiation capability of ASCs, as we have reported[14,28].

### Decreased ADCY3 in ASCs derived from donors with obesity
After systematically analyzing transcriptomic data, characterizing deregulated genes and signaling pathways, and specifying corresponding defective functionalities, we paid high attention to the two crucial regulators of ciliary biogenesis, CLUAP1/IFT38 and ADCY3. CLUAP1/IFT38, a key component of the IFT-B peripheral subcomplex, is essential for cilium assembly and maintenance[61]. *Cluap1*-deficient mice were not viable, and *Cluap*$^{-/-}$ mouse embryonic fibroblasts failed to form primary cilia and were defective in Hedgehog signaling[62,63]. To examine this regulator in more detail, we stained ASCs with the CLUAP1 antibody, acetylated α-tubulin, and DAPI (Fig. 5A) for further analysis. Microscopic analysis revealed that CLUAP1 localized to the base and the tip of the cilia, as reported[63]. Its protein intensity along the primary cilium was further evaluated via a line-scan analysis[14,64].

This evaluation found that the protein levels of axonemal CLUAP1 were reduced in obese ASCs at the base (Fig. 5A, B). Moreover, ADCY3, a transmembrane enzyme, is regulated by various GPCRs and catalyzes the synthesis of cAMP, a critical intracellular second messenger in signaling pathways that broadly impact cellular and ciliary functions[65,66]. Intriguingly, ADCY3 was enriched at primary cilia of ASCs, which was more obvious at ASCs derived from lean donors (Fig. 5C). Further line-scan analysis showed that ADCY3 protein levels in the ciliary region were significantly reduced along the entire cilium in obese ASCs (Fig. 5D), suggesting that diminished ADCY3 may contribute to cilium shortening in these cells. To test the functional involvement of ADCY3 in regulating ciliary length, lean ASCs were transfected with either control siRNA (sicon) or siRNA against ADCY3. Knockdown efficiency was confirmed by RT-PCR (Fig. 5G). Further analysis revealed that ADCY3 knockdown significantly reduced ciliary length (Fig. 5E, F), reaching levels comparable to those observed in ASCs from donors with obesity (Fig. 4B). These findings strengthen the involvement of ADCY3 in maintaining the length of primary cilia.

### TGFβ drives ciliary shortening and gene suppression similar to the obese phenotype
We further investigated potential mechanisms underlying impaired ciliation in ASCs from individuals with obesity. Members of the regulatory factor X (RFX) family are among the most well-established transcriptional regulators of primary cilium biogenesis and regulate target genes such as *IFT172*[67,68]. Based on the data from transcriptomic profiling, ASCs express the transcription factor genes *RFX1*, *RFX2*, and *RFX3* (Fig. 6A). Although not significantly altered in the transcriptome data, *RFX2* emerged as a promising candidate due to its significantly reduced expression in obese ASCs, as demonstrated by RT-PCR (Fig. 6B).

To examine whether the elevated TGFβ signaling observed in obese ASCs contributes to ciliary shortening and reduced expression of ciliary genes, we treated lean ASCsub cells with TGFβ for 48 h. This treatment led to a measurable reduction in ciliary length (Fig. 6C, D). In parallel, RT-PCR analysis showed that TGFβ significantly downregulated the expression of the ciliary transcription factor gene *RFX2*, the enzyme *ADCY3*, and two intraflagellar transport genes, *IFT88* and *IFT172*, which are essential for cilium assembly (Fig. 6E, F). To affirm treatment efficiency, we also assessed the expression of established TGFβ target genes, *COL1A1* and *SERPINE1*, which were significantly upregulated, confirming pathway activation (Fig. 6G). These findings indicate that TGFβ may promote ciliary shortening in obesity partly through the suppression of critical ciliary regulator genes, including *RFX2*, which may in turn lead to reduced expression of ADCY3, though further work is required to decipher how TGFβ reduces the expression of these important ciliary regulators.

### Modulation of adenylate cyclase activity affects cilium length and ASC motility
To further investigate the role of ADCY3 and cAMP signaling in ciliary regulation, ASCs from lean donors were treated with the ADCY inhibitor SQ22536, which blocks cAMP signaling[69], whereas ASCs from donors with obesity were incubated with an ADCY activator NKH477 (colforsin dapropate hydrochloride), a water-soluble forskolin derivate, leading to an increase in cAMP levels[70]. The effect of both compounds was proven by intracellular cAMP measurements via ELISA (Fig. 7A, B). To test the

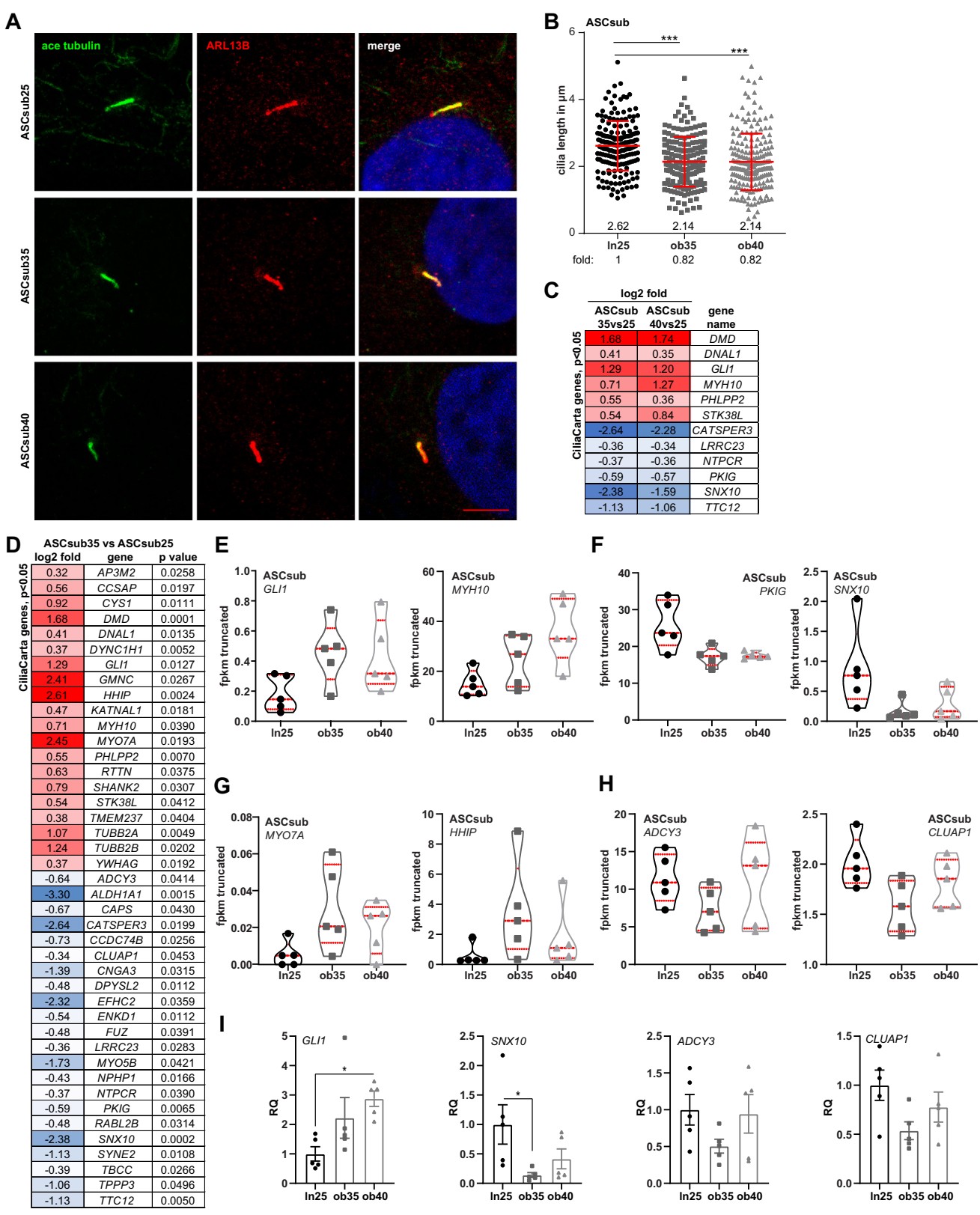

hypothesis that ADCY3 activity or cAMP availability affects primary cilium length in ASCs, we performed immunofluorescence staining with the ciliary markers acetylated α-tubulin and ARL13B to assess changes in cilium length (Fig. 7C, F). Primary cilia in ASCs, derived from lean donors and treated with two different concentrations of SQ22536 for 24 and 48 h, became shorter (Fig. 7D, E). Instead, ASCs derived from donors with a BMI above 35

and treated with NKH477 displayed up to 24% increase in cilium length (Fig. 7G, H). As these treatments did not influence cell viability (Fig. S3A), the findings strongly suggest that the deregulated ADCY3/cAMP signaling pathway contributes to shorter cilia in obese ASCs. As controls, lean ASCs were also treated with NKH477, and obese ASCs with SQ22536. Neither treatment affected cell viability (Fig. S3B). While activation of ADCY3 with

**Fig. 4 | Decreased cilium length in ASCs isolated from donors with obesity.**
**A** Subcutaneous ASCs (ASCsub) were stained for the ciliary markers acetylated α-tubulin (ace tubulin, green) and ARL13B (ADP-ribosylation factor-like GTPase 13B, red), and DNA (DAPI, blue). Representatives are shown. Scale: 5 μm. **B** Cilium lengths were measured and are presented as scatter dot plots (mean ± SD). The results are based on at least six individual samples per group (ASCsub25, n = 6 individual individual samples, 180 cilia; ASCsub35, n = 6, 178 cilia; and ASCsub40, n = 7, 208 cilia) and statistically analyzed. **C, D** The heatmaps depict significantly ($p < 0.05$) differentially expressed genes from CiliaCarta[49] in both ASC subgroups of donors with obesity, or in ASCsub35 vs. ASCsub 25, respectively. Significantly enriched genes have a red color code and reduced genes are encoded in blue. Red and blue colored genes are sorted alphabetically. **E–H** Truncated violin plots present selected ciliary genes, which are differentially expressed. Values reflect the fragments per kilobase per million mapped fragments (fpkm) of genes from the RNA-seq data. Each violin plot displays the median (central dashed line) and quartiles (upper and lower dotted lines). **I** Relative gene levels of *GLI1* (glioma-associated oncogene homolog 1), *SNX10* (sorting nexin 10), *ADCY3* (adenylate cyclase 3), and *CLUAP1* (clusterin associated protein 1) were corroborated with RT-PCR. The results are obtained from five individual samples in each group and presented as relative quantification (RQ) with SEM. *GAPDH* was used as endogenous control. Ordinary one-way ANOVA followed by Dunnett's multiple comparisons test was used to assess statistical significance relative to ln25 (**I**). For non-Gaussian data (**B, I**, *GLI1*), the Kruskal-Wallis test followed by Dunn's post hoc test was applied. *$p < 0.05$, ***$p < 0.001$.

NKH477 increased slightly the cilium length in lean ASCs (8%, Fig. S3C), inhibition with SQ22536 hardly changed it in obese ASCs (Fig. S3D).

The motility of MSCs/ASCs is vital for the development, maintenance, and repair of body-wide tissues[71], and the primary cilium is involved in cell motility and migration[72]. To further investigate the restoration of the cilium length on the migratory ability of ASCs, time-lapse microscopy with single cell tracking was performed with lean ASCs in absence or presence of the ADCY inhibitor SQ22536, and with obese ASCs without or with its activator NKH477 (Fig. 8A, B). The velocity of untreated obese ASCs (ob con) was significantly decreased compared to lean ASCs (ln DMSO) (Fig. 8C, 1st vs. 3rd row). Relative to lean ASCs, obese ASCs had a reduced velocity of 44% (0.322 ± 0.082 vs. 0.181 ± 0.064 μm/min), which was raised by 54% following the stimulation with NKH477 (0.278 ± 0.080 μm/min) (Fig. 8C, 4th row, ob NKH). As a further control, lean ASCs treated with the ADCY inhibitor SQ22536 (ln SQ) showed a reduction of 34% (0.212 ± 0.066 μm/min) (Fig. 8C, 2nd row), compared to its untreated control (ln DMSO). Similar results were observed for the accumulated distance, with 215 ± 55 for lean con, 121 ± 43 μm for obese con, 185 ± 52 μm for obASCs treated with NKH477, and 142 ± 44 μm for lnASCs with SQ22536 (Fig. 8D). The directionality was significantly increased by 10% in obASCs upon NKH477 treatment (Fig. 8E). As controls, treatment of lean ASCs with NKH477 resulted in only a slight, but non-significant, improvement in velocity and accumulated distance, whereas obese ASCs treated with SQ22536 showed a slight, yet non-significant, reduction in these parameters (Fig. S3E–H). These data highlight that reduced ADCY3 is responsible, at least partially, for shortened cilia and that its activation rescues the cilium length in obese ASCs and, consequently, the motility of obese ASCs.

## Discussion

We have previously reported that ASCs are impaired in various functions and properties during obesity[14,15]. The present study deepens the molecular understanding of the extensive transcriptomic and functional alterations of ASCs from donors with obesity. In particular, we show that ASCs isolated from obese donors exhibit an altered ECM and TGFβ gene profile, along with abnormal ciliary biogenesis and shortened primary cilia, impaired differentiation capacity, and reduced cell motility.

Interestingly, the observed altered pathways and phenotypes are intriguingly linked. Enhanced TGFβ release is well characterized in obesity[34,35]. In addition, obesity-associated ECM stiffness is thought to promote F-actin assembly, which, in turn, facilitates TGFβ release, activating TGFβ signaling[43]. On the other hand, actin polymerization and stress fiber formation are known to impair ciliogenesis[73]. Notably, TGFβ signaling inhibits adipogenesis by suppression of PPARγ[74]. Moreover, ciliary receptors are also critical for the activation of adipogenesis in preadipocytes[75] and functional cilia are necessary for adipogenic and osteogenic differentiation of MSCs and ASCs[14,56,57], whereas inhibition of ciliary elongation by IFT88 siRNA reduced the adipogenic differentiation capacity[56]. Functional ASCs require properly regulated ECM and ciliary biogenesis[9,15]. Notably, obesity changes the composition, structure, and function of the ECM of AT, which has been suggested to play a major role in defining the severity of obesity[76]. Therefore, the intertwined relationship between defective ECM, enhanced TGFβ signaling, and altered ciliary biogenesis could lead to impaired differentiation capacity of obese ASCs. In support of these observations, our results show that obesity leads to transcriptional alterations of ASCs, particularly genes involved in various aspects of the ECM and actin cytoskeleton. Both ASCsub35 and ASCsub40 abundantly express various collagen genes (*COL*). Interestingly, altered collagens contribute to AT dysfunction and/or metabolic diseases and are often highly expressed in obese tissue[30]. Increased expression of *COL4A1*, *COL5A2*, and *COL12A1* was also found in the transcriptomic analysis of subcutaneous AT of obese individuals[77]. The expressed collagens shifted from collagen types I, III, and V to the types IV and VI during the differentiation of 3T3-L1 preadipocytes into adipocytes[30,78]. The downregulation of FN-integrin α5 interactions has also been reported to be necessary for adipogenesis[12]. In fact, we show here an increased expression of *FN1* in obese ASCs with impaired adipocyte differentiation capability. Furthermore, the primary cilium and ciliary cAMP play a critical role in directing the cell fate of adipocyte progenitor cells in multiple AT types[79]. The cilium length of human ASCs/MSCs dynamically changed during adipogenic differentiation, likely due to ciliary trafficking of various receptors, while blocking this elongation impaired adipogenic induction[56,57]. These data underline the importance of the primary cilium in the adipogenic capacity and cell fate of ASCs.

Ciliopathies are often accompanied by obesity, and associated with fibrosis, ECM accumulation, and enhanced expression of TGFβ, suggesting that the impairment of primary cilia may be involved in the pathogenesis of obesity[80,81]. Obesity-associated factors shortens its length and impairs its functionalities in ASCs[14]. However, the molecular mechanisms are not entirely defined. It has been described that RFXs are well-established transcription factors of ciliary biogenesis[67,68]. We report here that *RFX2* is significantly downregulated in ASCs from donors with obesity, which may lead to reduced expression of ADCY3. Importantly, the treatment of lean ASCs with TGFβ reduces ciliary length as well as the gene expression of the transcription factor *RFX2*, its downstream genes *ADCY3*, and *IFT88* and *IFT172*, which are essential for cilium assembly. In line with our data, various components of the TGFβ signaling machinery have been reported to be present at the primary cilium[82]. In further support, TGFβ treatment resulted in reduced ciliary length by upregulating histone deacetylase 6 activity in human osteoblasts[83], or suppressed the levels of IFT88 mRNA and protein of murine chondrocytes[84]. These data clearly demonstrate that TGFβ is vital for ciliary biogenesis. Therefore, obesity-associated increase in TGFβ contribute to impaired primary cilia formation in obese ASCs, thereby compromising their functional properties.

*ADCY3* is downregulated in obese ASCs. It has been revealed that three types of adenylate cyclase localize to the primary cilium (ADCY3, ADCY5, and ADCY6)[85]. These enzymes catalyze the synthesis of the second messenger cAMP, which is a central player in ciliary signaling[86]. ADCY3 is localized at primary cilia of neurons[87], primary epithelial kidney cells[88], human fibroblasts or fibroblast-like synoviocytes[89,90], and MLO-Y4 osteocyte-like cells[91]. Distinct human *ADCY3* genetic variants have been linked to the pathogenesis of monogenic and polygenic obesity as well as type 2 diabetes[92]. Moreover, mice heterozygous and homozygous null for *Adcy3* exhibited increased susceptibility to obesity associated with insulin or leptin resistance[93,94]. Interestingly, the inhibition of *Adcy3* in neurons, and consequently ciliary signaling, increased the body weight of mice[95]. In line with

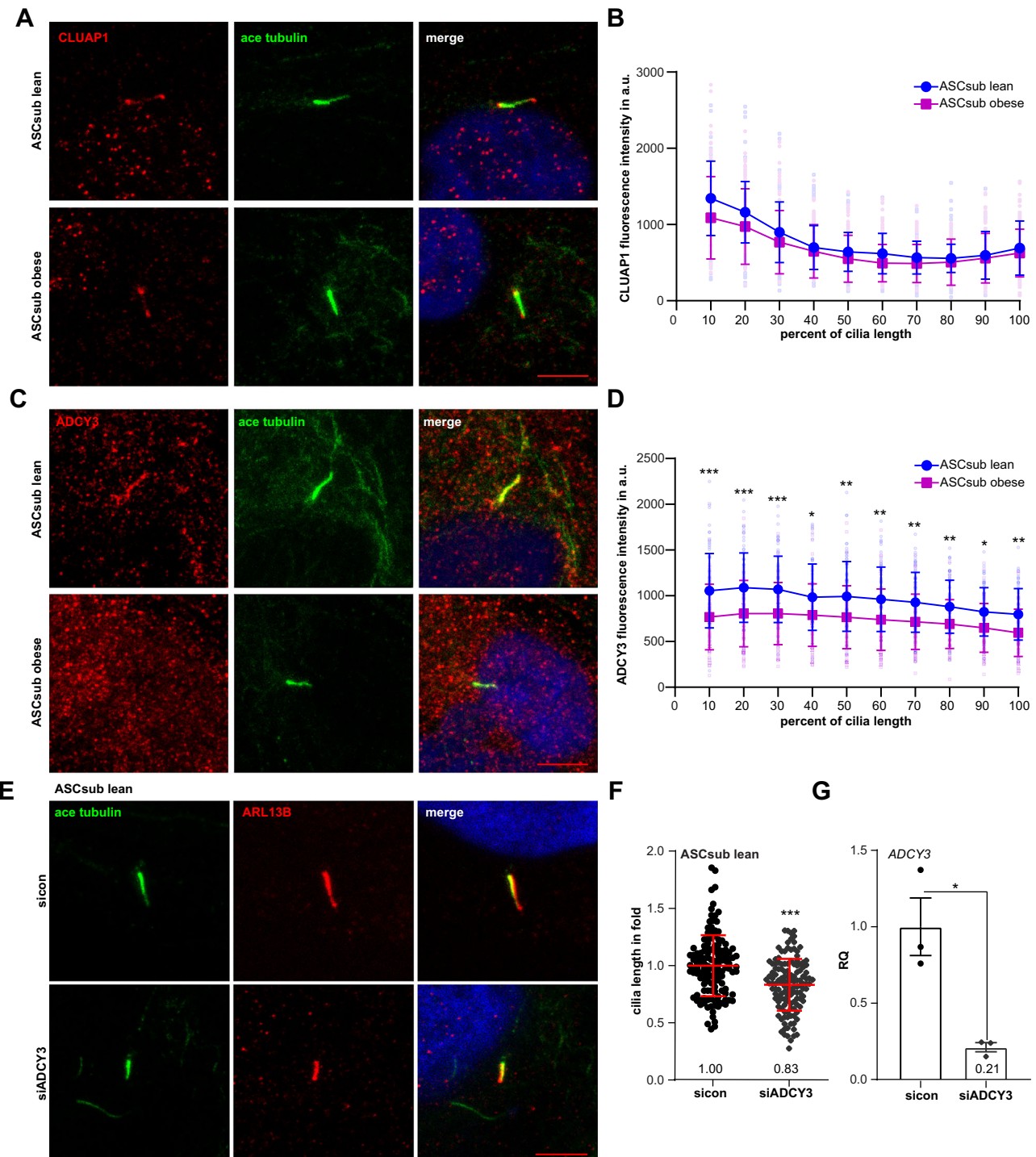

**Fig. 5 | Reduced ADCY3 intensity at the cilium of ASCs from donors with obesity.**
**A** Cells were stained for the ciliary markers clusterin associated protein 1 (CLUAP1, red) and acetylated α-tubulin (ace tubulin, green), and DNA (DAPI, blue). Representatives are shown for measurements of primary cilium staining of CLUAP1. Scale: 5 μm. **B** Line-scan analyses of axonemal CLUAP1 are shown for ASCs. Each point of the curve represents the mean fluorescence intensity ±SD and the results are based on three individual samples (n = 3, 73 cilia), normalized to the cilium length in percentage; a.u., arbitrary unit. **C** Cells were stained for the ciliary markers ADCY3 (adenylate cyclase 3, red) and acetylated α-tubulin (ace tubulin, green), and DNA (DAPI, blue). Representatives are shown for measurements of the primary cilium staining of ADCY3. Scale: 5 μm. **D** Line-scan analyses of axonemal ADCY3 are shown for ASCs. Each point of the curve represents the mean fluorescence intensity ±SD and the results are based on three individual samples (n = 3, 72 cilia), normalized to the cilium length (in percent); a.u., arbitrary unit. Statistical significance

was assessed using the Kruskal-Wallis test followed by Dunn's post hoc test to compare lean and obese groups at each 10% interval of relative ciliary length (0–100%). *$p < 0.05$, **$p < 0.01$, ***$p < 0.001$; ns, not significant. **E–G** Lean ASCs were treated with 30 nM of control siRNA (sicon) or with siRNA against ADCY3 for 48 h. **E** Primary cilia were stained for the ciliary marker acetylated α-tubulin (ace tubulin, green) and ARL13B (ADP-ribosylation factor-like GTPase 13B, red), and DNA (DAPI, blue). Representatives are shown. Scale: 5 μm. **F** The lengths of individual cilia were measured and are presented as scatter dot plots (mean ± SD). The results are based on three individual samples per group (n = 3, 135 cilia) and statistically analyzed. **G** Relative gene levels of *ADCY3* are shown. The results were obtained from three individual samples in each group and presented as relative quantification (RQ) with SEM. *GAPDH* served as endogenous control. Unpaired Mann–Whitney *U* test (**F**) or Student's *t* test was used (**G**). *$p < 0.05$, ***$p < 0.001$.

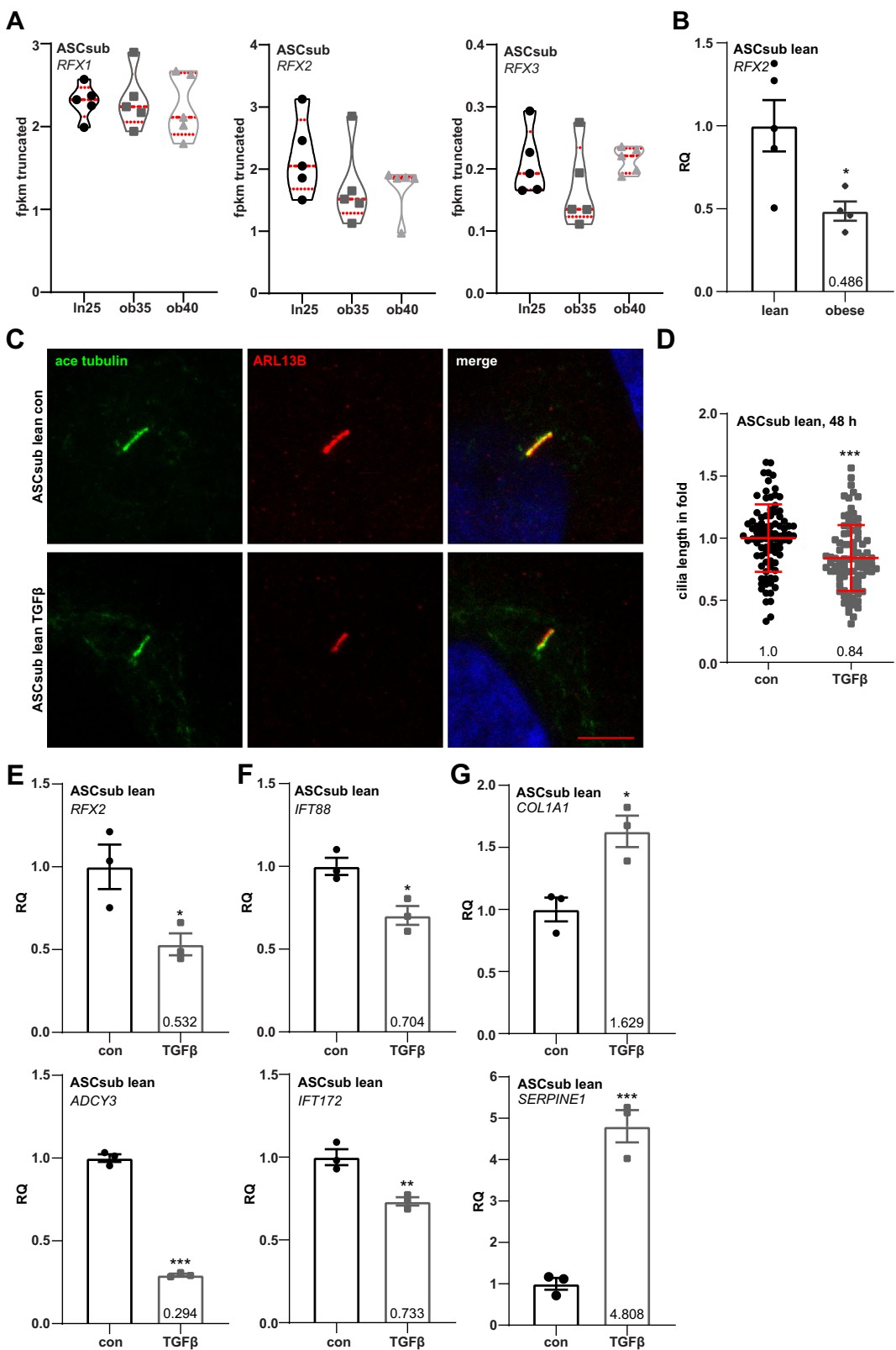

these observations, we report here that ADCY3 is localized at primary cilia of ASCs, its abundance is significantly reduced throughout the primary cilium in obese ASCs, and siRNA against ADCY3 reduced the cilium length of lean ASCs to a level comparable to that in obese ASCs. Altogether, our data clearly suggest that reduced *RFX2* and ADCY3 could contribute to

defective cilia of obese ASCs. These findings also highlight that the TGFβ-RFX-ADCY network is of importance in ciliary biogenesis.

Interestingly, it has been supposed that the stimulation of the *ADCY3* activity may be a useful treatment approach for obesity[65]. We demonstrate here that treatment with the ADCY3 activator NKH477 restores the cilium

**Fig. 6 | Transforming growth factor-β (TGFβ) drives ciliary shortening and gene suppression similar to the obese phenotype. A** Truncated violin plots of the members of the regulatory factor X (RFX) family (*RFX1*, *RFX2*, and *RFX3*). Values reflect the fragments per kilobase per million mapped fragments (fpkm) of genes from the RNA-seq data. Each violin plot displays the median (central dashed line) and quartiles (upper and lower dotted lines). **B** Relative gene levels of *RFX2* were corroborated with RT-PCR. The results are obtained from five individual samples for the lean and four individual samples of the obesity group, and presented as relative quantification (RQ) with SEM. *GAPDH* was used as endogenous control. **C–G** Treatment of lean ASCsub with TGFβ for 48 h. **C** Primary cilia of lean ASCsub were stained for the ciliary markers acetylated α-tubulin (ace tubulin, green) and ARL13B (ADP-ribosylation factor-like GTPase 13B, red), and DNA (DAPI, blue). Representatives are shown. Scale: 5 μm. **D** Cilium lengths were measured and are presented as scatter dot plots (mean ± SD). The results are based of three individual samples per group. **E–G** Relative gene levels of *RFX2*, *ADCY3* (adenylate cyclase 3), *IFT88* (intraflagellar transport), *IFT172*, *COL1A1* (collagen 1A1), and *SERPINE1* (serpin family E member 1) are shown. The results were obtained from three individual samples in each group and presented as relative quantification (RQ) with SEM. *GAPDH* served as endogenous control. Statistical significance was analyzed with the Student's *t* test. *$p < 0.05$, ***$p < 0.001$.

length of ASCs derived from donors with obesity, whereas reducing cAMP levels by inhibiting ADCY3 results in shorter cilia in lean ASCs. In support of our data, using optogenetic tools, Hansen et al. showed that an increase in ciliary cAMP lengthened the primary cilium of the mouse kidney cell line IMCD3[96]. In addition, Ansari et al. demonstrated that reduced ciliary cAMP production or *Adcy3* knockout led to primary cilium shortening in IMCD3 cells, which was restored after treatment with the cAMP analog dibutyryl-cAMP[97]. Our data, in combination with reported results, highlight that the ADCY3-cAMP signaling axis is a central component of ciliary signaling, which is disturbed in ASCs derived from donors with obesity.

The primary cilium is involved in the control of cell motility[80,98], in neuronal cells as well as non-neuronal, including endothelial cells, fibroblasts, and MSCs[66,99], especially ASCs[14,26]. Primary cilium signaling can influence the migratory response by affecting cytoskeletal reorganization, cell polarity, adhesion dynamics, directionality, and speed[98]. Studies show that the primary cilium senses and responds to ECM changes and that its formation, orientation, and length dynamics may act as a switch for regulating cellular responsiveness during migration[98]. Interestingly, genetic ablation of the primary cilium, or knockdown of *Adcy3* caused migratory defects of neurons with reduced migratory speed[99]. We show here that ASCs derived from donors with obesity exhibit shorter cilia, reduced levels of *RFX2* and ADCY3, and impaired cell migration. These defects can be partially remedied by targeting ADCY3 and enhancing intracellular cAMP levels. Treatment with an ADCY activator improves the velocity, accumulated distance, and directionality of impaired ASCs from donors with obesity.

## Conclusions

This study highlights that obesity induces transcriptional reprogramming of ASCs, leading to alterations in ECM composition and ciliary biogenesis. The impairment of primary cilia in obese ASCs, particularly through down-regulation of key regulators like ADCY3, probably through the transcription factor *RFX2* that is regulated by TGFβ, may contribute to defective differentiation capacity and cell motility, potentially fueling the progression of obesity (Fig. 8F). Moreover, the reduced cAMP signaling in obese ASCs, which can be partially restored through the activation of the ADCY3-cAMP signaling axis, further underscores the critical role of primary cilia in regulating ASC function. Although further investigation is needed, these findings suggest that obesity-induced changes in ECM, TGFβ signaling, ciliary biogenesis, and cAMP signaling may impair ASC differentiation and migration, highlighting potential therapeutic targets for obesity-related disorders.

## Materials and methods
### Ethics approval and consent to participate
This work was approved by the Ethics Committee of the Johann Wolfgang-Goethe University Hospital Frankfurt (reference number: 375/11, title: Comparison of metabolic pathways in normal and preeclampsia pregnancies, approved newly on 23th November 2021) and informed written consent was obtained from all donors. The Ethics Committee acts in accordance with the Helsinki Declaration. All ethical regulations relevant to human research participants were followed.

### ASC isolation, treatment, transfection, and differentiation
Subcutaneous AT samples were taken from women undergoing cesarean section. Previous studies have shown that the reproductive state has minimal impact on the differentiation capacity of ASCs[100]. All participants showed no evidence of impaired glucose tolerance, as confirmed by the 50 g oral glucose tolerance test (oGTT50) and, if indicated, an oGTT75 or hemoglobin A1c. Participant information is listed in Table 1.

ASCs were isolated as reported[25,101]. The isolated cells were resuspended in DMEM with pyruvate, 4.5 g/L D-glucose and L-glutamine (Gibco, Carlsbad) containing 20% fetal bovine serum, 1% penicillin/streptomycin (Sigma-Aldrich), and 1 μg/ml amphotericin-B (Sigma-Aldrich), and cultured under standard cell culture conditions. All ASCs were routinely tested for mycoplasma (Minerva Biolabs GmbH).

ASCs were treated with the ADCY inhibitor SQ22536 (sc-201572, Santa Cruz Biotechnology, Inc., Heidelberg), dissolved in dimethyl sulfoxide (DMSO, Thermo Fisher Scientific), the ADCY activator NKH477 (sc-204130, Santa Cruz), or TGFβ (#100-21-2UG, PeproTech). Concentrations and time points are indicated in each experiment. To match the maximal concentration of SQ22536, DMSO was used at a final concentration of 0.05% (v/v).

ASCs were transfected with 30 nM of control small interfering RNA (AllStars Negative Control siRNA, Qiagen, Hilden, #1027281) or siRNA targeting ADCY3 (Santa Cruz Biotechnology, Inc., sc-29600) using the 4D-Nucleofector™ Core Unit with X Unit (Amaxa™ P1 Primary Cell 4d-Nucleofector™ Kit, #V4XP-102) with the program DO-101 (Lonza, Maastricht).

Adipogenic differentiation of ASCs was performed as reported[25]. Briefly, ASCs were cultured with optimized differentiation medium for the generation of adipocytes (StemMACS™ AdipoDiff Media, #130-091-677, Miltenyi Biotec, Gladbach) up to 14 days. Cells were fixed and stained with BODIPY™ 493/503 (D3922, Invitrogen) and phalloidin (Phalloidin-Atto 550, Sigma-Aldrich), or with Oil Red O (Sigma-Aldrich) and hematoxylin (Sigma-Aldrich) to analyze adipogenic differentiation. For osteogenic and chondrogenic differentiation, ASCs were incubated with StemMACS™ OsteoDiff Media (#130-091-678) or ChondroDiff Media (#130-091-679, Miltenyi Biotec) for 21 days[28].

### Surface marker measurement
The cell surface markers of ASCs were measured via FACS with FACSCalibur™ (BD Biosciences, Heidelberg). Cells were trypsinized with 0.25% trypsin (Sigma-Aldrich), harvested, and fixed for 15 min with ice-cold 2% paraformaldehyde. After washing twice with PBS, the cells were stained with the following antibodies: PE-conjugated anti-human cluster of differentiation 44 (CD44) (#130-113-342, clone REA690, lot 5240507442, MACS Miltenyi Biotec, Bergisch Gladbach), PE-conjugated anti-human CD73 (#550257, lot 4149735, BD-Pharmingen, Heidelberg), FITC-CD90 (#11-0909-42, clone eBio5E10, lot 2648893, eBioscience, Frankfurt), PE-conjugated anti-human CD105 (#323206, clone 43A3, lot B418585, BioLegend, San Diego), PerCP-Cy5.5-conjugated anti-human CD14 (#45-0149-42, clone 61D3, lot 2785933, eBioscience), APC-conjugated anti-human CD31 (#17-0319-42, clone WM-59, lot 2702043, eBioscience), and FITC-conjugated anti-human CD34 (#343504, clone 581, lot B407007, eBioscience). Non-stained ASCs were used as negative controls. All ASCs were characterized for the positive (CD44, CD73, CD90, CD105) and the negative markers (CD14, CD31, CD34), as listed in Table 2. An example for the gating strategy is provided in Supplementary Information (Fig. S4). The passages P3 to P8 of isolated ASCs were used for experiments.

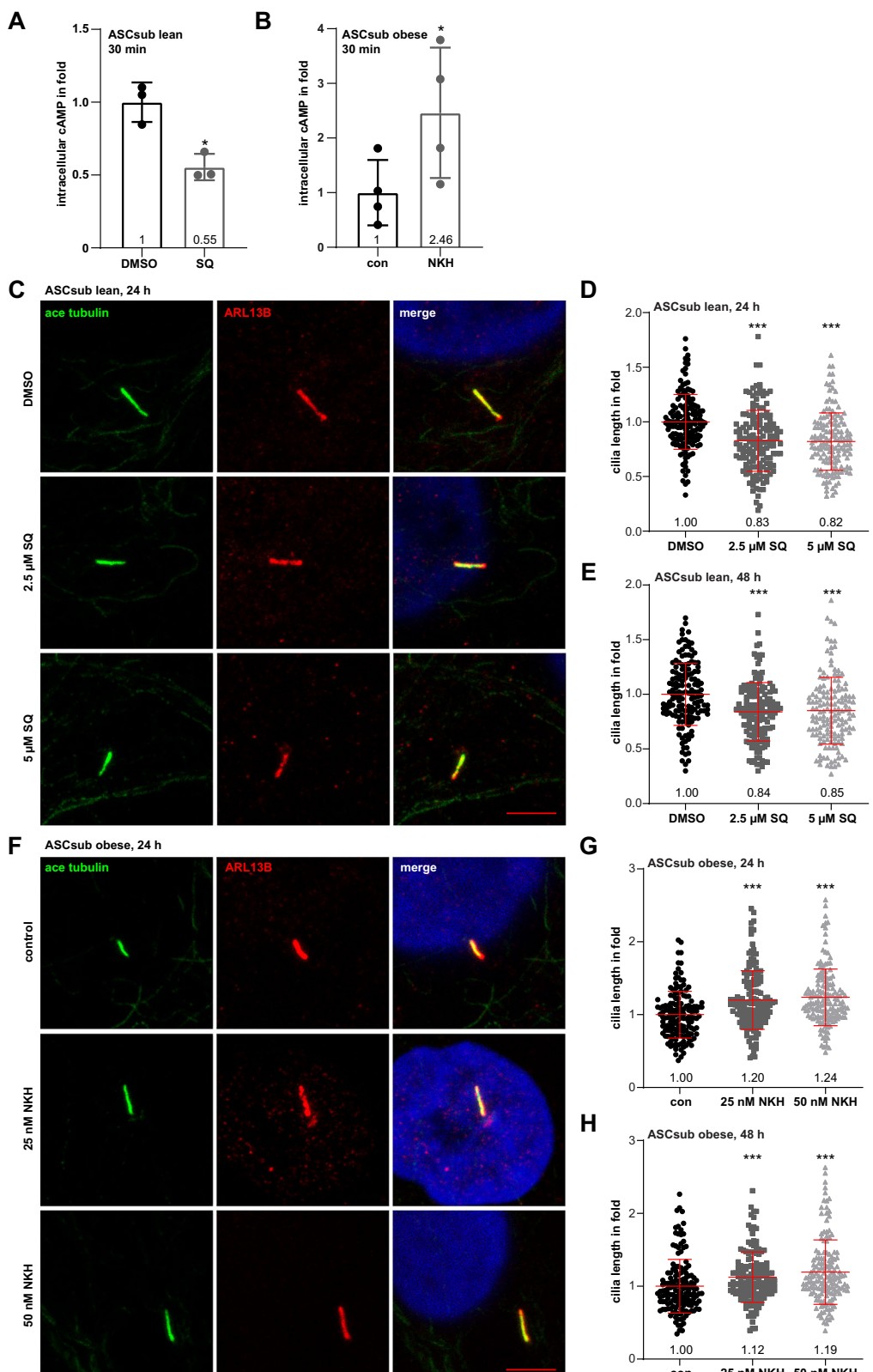

**Fig. 7 | Alterations in cilium length depend on ADCY3/cAMP signaling. A**, **B** The effect of the ADCY inhibitor SQ22536 (SQ) or its activator NKH477 (NKH) was proven by the measurement of intracellular concentrations of acetylated cAMP via a direct cAMP ELISA. The results are presented as mean ± SD. Student's *t* test was used. * *p* < 0.05. ASCs were treated with the ADCY inhibitor SQ22536 (SQ) (**C–E**) or its activator NKH477 (NKH) (**F–H**) for 24 or 48 h with indicated concentrations. Primary cilia of ASCsub were stained for the ciliary markers acetylated α-tubulin (ace tubulin, green) and ARL13B (ADP-ribosylation factor-like GTPase 13B, red), and DNA (DAPI, blue). Representatives are shown. Scale: 5 μm (**C**, **F**). Cilium lengths were measured and are presented as scatter dot plots (mean ± SD) (**D**, **E** and **G**, **H**). The results are based on three individual samples per group (n = 3, 150 cilia) and statistical significance was analyzed with the Kruskal-Wallis test followed by Dunn's post hoc test. ****p* < 0.001. Abbreviations: ADCY3 adenylate cyclase 3, cAMP cyclic adenosine monophosphate, con control, DMSO dimethyl sulfoxide, ELISA enzyme-linked immunosorbent assay, min minutes.

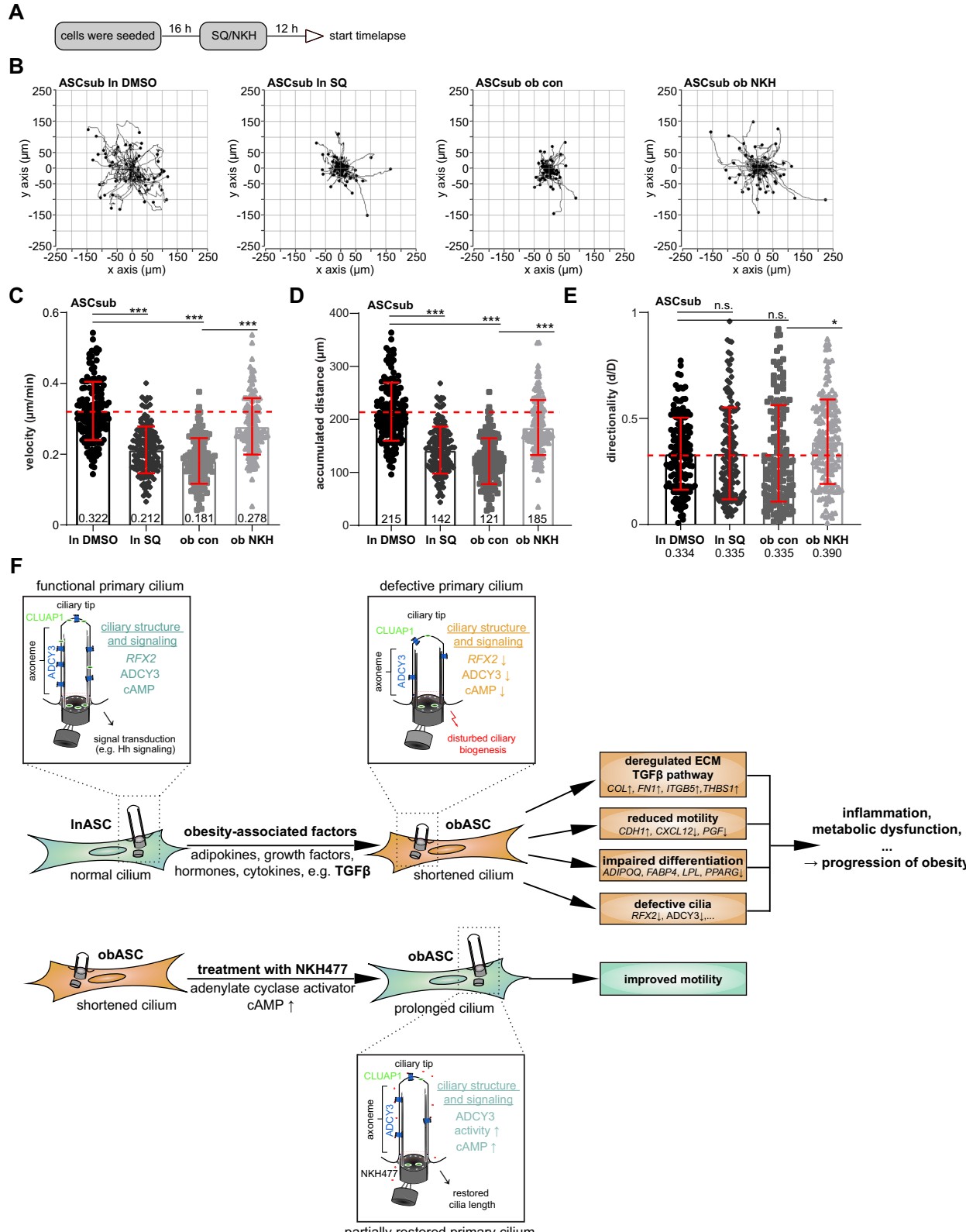

## RNA extraction, transcriptomic analysis (RNA-seq), and real-time PCR (RT-PCR)

For transcriptomic analysis, total RNAs were extracted from each ASC sample (cells at P4) of five different donors per condition with EXTRACTME TOTAL RNA Kit (#EM09.2.-250, 7Bioscience, Neuenburg). A total amount of 1.5 µg RNA per sample was forwarded to Novogene Co., Ltd. (London) and used as starting material for RNA sample preparation[28]. The Venn diagram, volcano plots, violin plots, and pathway analyses were either generated by Novogene Co. Ltd. or by re-analyzing the data using Microsoft® Excel® (Redmond, Washington) and GraphPad Prism 8 or 10

**Fig. 8 | Reactivation of ADCY3 signaling enhances the motility of ASCs from donors with obesity.** Analysis of cell motility. **A** Experimental schedule of treatment with the ADCY inhibitor SQ22536 (SQ) or its activator NKH477 (NKH). **B** Representative trajectories are depicted for individual cells (n = 1, 50 cells in each group). The velocity (**C**), the accumulated distance (**D**), and the directionality (**E**) of ASC motility are shown as mean ± SD (n = 3, 150 cells pooled from three experiments). Statistical significance was analyzed with the Kruskal-Wallis test followed by Dunn's multiple comparison test. *$p < 0.05$; ***$p < 0.001$; n.s., not significant. **F** Schematic illustration of the proposed working model. Obesity-associated factors, such as TGFβ, are able to shortened primary cilia in ASCs associated with

deregulated pathways and genes/proteins, fueling inflammation and metabolic dysfunction and facilitating the progression of obesity. Treatment with NHK477 reactivates the ADCY3-cAMP signaling axis, prolongs the primary cilium, improves cell motility, and thus possibly prevents the progression of obesity. Abbreviations: ADCY3 adenylate cyclase 3, ADIPOQ adiponectin, cAMP cyclic adenosine monophosphate, CDH1 cadherin-1, COL collagen, CXCL12 CXC motif chemokine 12, DMSO dimethyl sulfoxide, FABP4 fatty acid-binding protein 4, FN1 fibronectin, HH hedgehog, ITGB5 integrin beta 5, ln lean, LPL lipoprotein lipase, ob obese, PGF placental growth factor, PPARG peroxisome proliferator-activated receptor gamma, TGFβ transforming growth factor-β, THBS1 thrombospondin 1.

## Table 1 | Clinical information of donors

| group name | n | age (years) | gestational age (weeks) | pre-pregnancy body mass index (BMI) |
|---|---|---|---|---|
| ASCsub25/ln25 | 5 | 31.6 ± 4.3 | 40.2 ± 0.8 | 21.0 ± 2.0 |
| ASCsub35/ob35 | 5 | 33.2 ± 4.0 | 40.8 ± 1.3 | 37.0 ± 0.8 |
| ASCsub40/ob40 | 5 | 35.6 ± 2.9 | 39.6 ± 1.8 | 42.7 ± 1.6 |

Mean value ± standard deviation (SD) is shown.

## Table 2 | Cell surface markers of ASCs

| ASCsub | CD44 | CD73 | CD90 | CD105 | CD14 | CD31 | CD34 |
|---|---|---|---|---|---|---|---|
| lean 1 | 99.77 | 98.93 | 95.71 | 99.65 | 0.97 | 2.54 | 2.54 |
| lean 2 | 99.31 | 98.88 | 99.03 | 99.84 | 0.13 | 1.05 | 0.17 |
| lean 3 | 99.94 | 99.81 | 99.44 | 99.99 | 0.02 | 1.38 | 1.72 |
| lean 4 | 99.28 | 99.81 | 99.59 | 99.92 | 4.66 | 0.18 | 0.32 |
| lean 5 | 99.58 | 97.93 | 98.89 | 98.85 | 0.11 | 0.74 | 1.39 |
| mean ± SD | 99.6 ± 0.3 | 99.1 ± 0.8 | 98.5 ± 1.6 | 99.7 ± 0.5 | 1.2 ± 2.0 | 1.2 ± 0.9 | 1.2 ± 1.0 |
| obese 1 | 99.85 | 99.26 | 99.49 | 99.76 | 3.36 | 8.44 | 1.45 |
| obese 2 | 99.31 | 99.5 | 98.9 | 99.67 | 2.59 | 5.33 | 14.73 |
| obese 3 | 99.78 | 99.92 | 99.67 | 99.88 | 13.56 | 6.06 | 3.53 |
| obese 4 | 99.53 | 99.77 | 88.63 | 99.01 | 1.86 | 0.03 | 0.09 |
| obese 5 | 99.96 | 99.95 | 99.62 | 100 | 2.8 | 10.77 | 12.51 |
| mean ± SD | 99.7 ± 0.3 | 99.7 ± 0.3 | 97.3 ± 4.8 | 99.7 ± 0.4 | 4.8 ± 4.9 | 6.1 ± 4.0 | 6.5 ± 6.7 |

Exemplary results of FACS analyses. Mean value ± standard deviation (SD) in percentage is shown.
*CD* cluster of differentiation.

(GraphPad software Inc., San Diego). The Revigo tool[102] was used to reduce redundancy among GO term analyses, where a selection of significantly enriched pathways are shown. Additional Venn diagrams (Fig. S1A, B) were performed with the web-based tool InteractiVenn[29]. CiliaCarta was used to identify known ciliary genes[49].

For RT-PCR, reverse transcription was done using GoScript™ Reverse Transcription Mix, Random Primers (#A2801, Promega, Madison). The probes for *ADCY3* (Hs01086502_m1), *ADIPOQ* (Hs00605917_m1), *CDH1* (Hs01023895_m1), *CLUAP1* (Hs01032009_m1), *COL1A1* (Hs00164004_m1), *COL11A1* (Hs01097664_m1), *COL5A2* (Hs00893878_m1), *CXCL12* (Hs00171022_m1), *FABP4* (Hs01086177_m1), *GAPDH* (Hs02758991_g1), *GAS6* (Hs01090305_m1), *GLI1* (Hs00171790_m1), *IFT88* (Hs00901755_m1), *IFT172* (Hs00404485_m1), *LPL* (Hs00173425_m1), *PGF* (Hs00182176_m1), *PPARG* (Hs01115513_m1), *RFX2* (Hs01100925_m1), *RUNX2* (Hs01047973_m1), *SERPINE1* (Hs00167155_m1), *SNX10* (Hs00203362_m1), and *SOX9* (Hs00165814_m1) were from Thermo Fisher Scientific (Dreieich). RT-PCR was performed with a QuantStudio 3 (Applied Biosystems by Thermo Fisher Scientific). All results were shown as relative quantification (RQ)[103,104]. RQ values were calculated using the ΔΔCt method. Mean and standard error of the mean (SEM) were calculated from biological replicates based on the individual RQ values (n = 3 to 5).

### Indirect immunofluorescence staining, microcopy, and signal intensity measurement

Indirect immunofluorescence staining and microscopic evaluation were performed[105,106]. Cells were fixed with 4% paraformaldehyde containing 0.2% Triton X-100 for 15 min at room temperature. To evaluate the cilium length, and the relative intensity of ciliary proteins, ASCs were stained with mouse monoclonal antibody targeting acetylated α-tubulin (#T6793, clone 6-11B-1, lot 0000298764, Sigma-Aldrich), rabbit polyclonal antibody against ARL13B (#17711-1-1AP, lot 00142161, Proteintech, Herford), rabbit polyclonal ADCY3 (#19492-1-AP, lot 00046273, Proteintech), and rabbit polyclonal CLUAP1 (#17470-1-AP, lot 00025316, Proteintech). FITC- and Cy3-conjugated secondary antibodies (Jackson ImmunoResearch, Pennsylvania) and DAPI (4', 6-diamidino-2- phenylindole-dihydrochloride, Roche, Mannheim) were used. Confocal laser scanning microscopy (CLSM, Leica, Wetzlar) was conducted using a Leica CTR 6500 system, employing Z-stack imaging with a HCXPI APO CS 63.0×/1.4 oil-immersion objective. Z-stacks were acquired at 0.5 μm intervals. Representative images were generated by overlaying individual confocal Z-sections utilizing maximum intensity projection. All experiments, unless otherwise indicated, were independently performed with ASCs isolated from at least three different donors. Relative fluorescence intensities of ADCY3 and CLUAP1 were measured using line-scan-based analysis with

ImageJ and the ImageJ plugin Plot Roi Profile (National Institutes of Health, Bethesda)[14,64]. The mean values of 72 or 73 cilia from three individual ASC samples in each group were obtained within the intervals and were plotted to GraphPad Prism 8 (GraphPad software Inc.).

## cAMP ELISA

Cells were treated with 100 μM of SQ22536 or 50 nM of NKH477 for 30 min, respectively. cAMP (3',5'-cyclic adenosine monophosphate) was extracted from ASCs using 0.1 M HCl for 20 min. The intracellular concentrations of acetylated cAMP in the lysates were then determined using a direct cAMP ELISA kit (#ADI-901-066A, Enzo Life Sciences GmbH, Lörrach).

## Cell viability assay

Cell viability was assessed using the CellTiter-Blue® assay (#G808B, Promega, Walldorf) following the manufacturer's protocol.

## Cell motility evaluation via time-lapse microscopy

Cells were seeded into 24-well plates with a low confluency (30-40%) and were imaged at 5-min time intervals for 13 h. Time-lapse imaging was performed with an AxioObserver.Z1 microscope (Zeiss, Göttingen, Germany), imaged with an AxioCam MRc camera (Zeiss, Göttingen, Germany) provided with an environmental chamber to maintain proper conditions (37 °C, 5% $CO_2$)[101]. Time-lapse movies were analyzed using ImageJ 1.49i software (National Institutes of Health) with the manual tracking plugin. Tracks were derived from raw data points, and the velocity, accumulated distance, and directionality were calculated using the Chemotaxis and Migration Tool (Ibidi GmbH, Gräfelfing), and depicted by GraphPad Prism 8 (GraphPad software Inc.). Fifty random cells per experiment were analyzed and independently performed with ASCs isolated from three individual samples.

## Statistics and reproducibility

Outliers were identified using Grubbs' test (GraphPad QuickCalcs, San Diego). Statistical analyses were performed using GraphPad Prism 8 (GraphPad software Inc.). The normality of data distribution was assessed using the Shapiro-Wilk test. For comparisons between two groups, an unpaired Student's $t$ test was applied for normally distributed data, and the Mann–Whitney $U$ test for non-normally distributed data. To compare three or more groups, one-way ANOVA was performed, followed by multiple comparisons tests: Tukey's (to compare all groups), Dunnett's (to compare all groups versus a control), or Sidak's (to compare selected groups). If the assumption of homogeneity of variances was violated (Brown-Forsythe test), Welch's ANOVA followed by Dunnett's T3 multiple comparisons test was used. For non-normally distributed data, the Kruskal-Wallis test followed by Dunn's multiple comparisons test was applied. A $p < 0.05$ was considered statistically significant. All experiments were performed using ASCs from independent donors, with n representing the number of individual patients from whom cells were isolated. Unless otherwise specified, experiments were conducted with $n = 3$. For transcriptomic analysis, ASCs from $n = 5$ individual samples were used. Exact n numbers are indicated in the corresponding figure legends.

## Reporting summary

Further information on research design is available in the Nature Portfolio Reporting Summary linked to this article.

## Data availability

The RNA sequencing data have been deposited in the NCBI Gene Expression Omnibus (GEO) under accession number GSE308101. The source data behind the graphs in the paper are included in the Supplementary Data 1 file. All other data supporting the findings of this study are available from the corresponding author upon request.

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

## Acknowledgements

This work was funded by the German Research Foundation (Deutsche Forschungsgemeinschaft (DFG), project number 438690235 to J.Y.). We are grateful to our donors and our clinical team for making this study possible. We thank Drs. He and Anderson, Sloan-Kettering Institute, for kindly providing us the ImageJ plug-in Plot Roi Profile for analyzing the fluorescence intensity in cilia. We sincerely thank Mr. Dominic Menger (Georg-Speyer-

Haus, Institute for Tumor Biology and Experimental Therapy, Frankfurt) for providing access to the confocal laser scanning microscope (CLSM).

## Author contributions

N.N.K. designed and performed experiments, analyzed data, and draw, modified and improved the manuscript. A.F. performed experiments, analyzed the data and modified the manuscript. A.R. analyzed data and did critical reading. A.E.H. has acquired and informed donors. E.S. collected the samples. F.L. conceived the project. J.Y. conceived the project and draw the first manuscript draft and did critical reading. All authors read and approved the final manuscript.

## Funding

## Competing interests

The authors declare no competing interests.

## AI disclouser

The authors declare that they have not use AI-generated work in this manuscript. During the revision process of this work, the authors used OpenAI's language model (ChatGPT) to improve the readability and clarity of the manuscript. The authors subsequently reviewed and edited the content as necessary and take full responsibility for the final version of the manuscript.
