## [Transparent Peer Review file · Communications Biology]

In-Depth Analysis of Obesity-Associated Changes in Adipose Tissue-Derived Mesenchymal Stromal/Stem Cells and Primary Cilia Function

Corresponding Author: Dr Nina-Naomi Kreis

Version 0:

Reviewer comments:

Reviewer #1

(Remarks to the Author)

Summary

In this manuscript entitled "In-Depth Analysis of Obesity-Associated Changes in Adipose Tissue-Derived Mesenchymal Stromal/Stem Cells and Primary Cilia Function", Kreis et al. describe the transcriptomic profile occurring in primary adipose tissue progenitors of individual with and without obesity. They first put the focus on alterations in the expression of genes involved in the extracellular matrix and adhesion molecule compositions, and their master regulator, the TGFb signalling pathway. In the second part of the manuscript, the authors switch the focus on primary cilia genes and cilia structure and assembly, demonstrating alterations in adipose tissue progenitors during obesity. Of note, the length and mobility of primary cilia are reduced, and the authors propose that alteration in the ADCY3-cAMP signalling cascade mediates these alterations.

Comments:

- 1) In Figure 1, panels A and C do not provide information regarding the dataset, this should be placed in supplementary. Line 205, the term "relevant" is used, what does "relevant" mean?
- 2) In Figure 1, the authors should better characterise the pathway analysis in correcting for redundancy in the list of genes contained in each pathways, by using PathFinder or an equivalent tool that corrects for such an overlap in terms. For example, in 1D, the authors state that multiple pathways linked to the ECM are identified "including genes responsible for the ECM in general (69 genes), proteinaceous ECM (56 genes), extracellular structure organisation (54 genes), and ECM organisation (47 genes)." It is very likely that the gene list of the different "pathways" overlap and this wrongfully emphasise the data, please show a Venn diagram of the common genes belonging to multiple pathways. Same for 1E, F and G. It is very likely that TGFb related pathways will be identified, as it is well described as a regulator of the ECM and is increased in adipose tissue during obesity, rather than manually isolating the term and depicting a few genes.
- 3) Figures 2, 3 and 4 are an overuse of transcriptomic data. The tables and volcano plot are fully redundant in the information provided to the reader. The qPCR validation should, at best, be provided as supplementary data. Figures 2, 3 and 4 are an extension of a currently incomplete Figure 1, and simply illustrate genes belonging to the pathways identified in Figure 1. Additionally, an increase in ECM production and TGFb signalling in adipocyte progenitors during obesity is already well characterised in both mice and humans (PMIDs : 22355352, 17065356, 18945811, 36617688, 30530781, 16253647, 10923640, 23449528). As a result, the main conclusions from the paragraphs 1 and 2 are not only redundant, but also only focus on known alterations during obesity.
- 4) Figure S2, an alteration of the adipogenic potential of AT progenitors from obese individuals compared to lean is already well described, especially through increased TGFb signaling (PMID : 30610207). Here the authors show quite a strong impairment, which might not reflect the reality of the pathophysiology of obesity. It would be reassuring to see data of differentiation using more conventional differentiation protocols with priming.

5) In Figure 7, what is the rationale between treating each group with only 1 compound? Do the authors claim that progenitors from obese individuals have reduced cAMP signalling, and this affects cilia length? As this is not demonstrated, they should treat both groups with both compounds and assess for cilia length (Lean CTL vs SQ vs NKH, and obese CTL vs NQ vs NKH).

6) Same in Figure 8, it would be easier to draw conclusions from the mobility measures if both lean and obese cells were treated with the 2 compounds.

7) The authors do not address whether improving ciliation is beneficial for adipogenesis, and thus the statement "The data suggest that impaired primary cilia in ASCs contribute to defective adipogenic differentiation and cell motility" should not be made. The authors should demonstrate that the compounds they use (SQ and NKH) can effectively influence adipogenesis. Furthermore, a proper demonstration can only be made by combining the treatments with a ADCY3 knock-down.

8) The authors already published that ciliation is impaired in adipose progenitors from individuals with obesity (Ref 14). They propose that decreased gene expression of the ciliary component shown here is the underlying mechanism. Gene expression alone is not sufficient to draw functional conclusions. Is reduced gene expression alone sufficient to reduce cilia formation? What is the mechanism driving this decrease in gene expression of ciliary components? Did the authors measure the protein and activation level of major transcription factors involved in cilium biogenesis such as the RFX family of FOXJ1?

9) Throughout the manuscript, there is an obvious attempt to justify the choice of genes plotted from the transcriptomic dataset. This leads to an overwhelming number of references and data for the reader to process, which dilute the main message rather than supporting the authors statements and make the manuscript difficult to read. The authors should narrow their writing and refer to relevant literature only when needed.

10) Statistics : Comparison of 3 or more groups can not be tested through multiple Student t-test nor Mann-Whitney. This is an abuse of statistical power. Please use ANOVA or Kruskal-Wallis depending on Gaussian distribution, combined with a post-hoc test for Fig2.E, Fig3.F, Fig4.E, Fig5.B, Fig7.E, F, H and I, and Fig8.C, D and E. Fig.6B and D, and Fig.S2E, F, G and H upper panels should be tested with a 2-Way ANOVA combined with post-hoc tests.

Reviewer #2

(Remarks to the Author)

• Brief Summary of the Manuscript

The manuscript reports novel findings regarding obesity and adipogenic differentiation, a topic of global pertinence. The authors explore primary ciliary dysfunction and its potential implications in obesity, particularly in studying adipose tissue-derived mesenchymal stromal/stem cells (ASCs), a relatively underexplored area of research.

The authors obtained ASCs from women undergoing cesarean section, categorizing them into three groups based on body mass index (BMI) for comparison: lean (BMI < 25, ASCsub25), obese 35 (BMI ≥ 35, ASCsub35), and obese 40 (BMI ≥ 40, ASCsub40). Utilizing transcriptomic gene analysis, real-time PCR, and immunofluorescence, the authors report that obesity alters the transcriptomic profile of ASCs, particularly affecting genes and pathways related to the extracellular matrix (ECM), transforming growth factor β (TGF β) signaling, cell motility, and adipogenic differentiation. They also indicate that genes involved in ciliary biogenesis are deregulated, particularly clusterin-associated protein 1 (CLUAP1) and adenylate cyclase 3 (ADCY3). Furthermore, the authors report that ASCs treated with an ADCY activator have a longer cilium and improved cell motility.

The authors conclude that defective ASCs may contribute to the development of obesity and that their restoration could present potential therapeutic targets for obesity-related disorders.

• Overall Impression of the Work

The study addresses a relevant topic in primary cilia dysfunction in ASCs. However, there are some aspects of the study and methodology that require further clarification from the authors.

• Specific Comments

1. There are inconsistencies in the use and definition of abbreviations throughout the manuscript. For example, 'ECM' is defined only in the abstract and not in the main text, whereas 'AT' is defined in the abstract and introduction. Additionally, both 'PPAR γ ' and 'PPARG' are used inconsistently. Review the manuscript for consistent and appropriate use of abbreviations.

2. In Table 2, the values are presented without specifying the corresponding units (e.g., percentage, arbitrary units, or mean fluorescence intensity...). Please clarify and include the appropriate units.

3. The manuscript does not provide sufficient detail regarding the clinical background of the donors from whom ASCs were isolated. It would be helpful to clarify whether the donors were otherwise healthy or had any conditions potentially associated with inflammation. For example, were glucose levels assessed? As hyperglycemia during pregnancy is relatively

common and may influence inflammatory status, such information could help interpret the data more accurately.

4. It is not clear whether the authors tested the viability of ASCs upon treatment with SQ22536 and NKH477 in their model. In addition, the final concentration of DMSO should be reported, as DMSO can interfere with multiple cellular pathways.

5. Approximately half of the figures in the manuscript are dedicated to transcriptomic analyses, many of which consist of violin plots. The high number of similar plots makes the figures visually dense. The authors may consider simplifying the presentation - for example, by transferring some of the detailed plots to the Supplementary Material.

Reviewer #3

(Remarks to the Author)

This study systematically investigates the impact of obesity on the functionality of adipose-derived mesenchymal stromal/stem cells (ASCs), focusing on extracellular matrix (ECM) remodeling, TGF β signaling, ciliary dysfunction, and cell motility. By integrating transcriptomic analysis, functional experiments, and pharmacological interventions, the authors reveal that obesity downregulates ADCY3/CLUAP1, leading to shortened cilia and functional impairment, thereby affecting ASC differentiation and migration. This study is interesting with clear logic, however, several critical issues require clarification to strengthen methodological rigor and mechanistic interpretation.

1. Why was subcutaneous fat from women undergoing cesarean section selected? Is the BMI measured during the third trimester or pre-pregnancy? This distinction is critical because weight gain during pregnancy is a normal physiological process, whereas pre-pregnancy BMI more accurately reflects an individual's basal metabolic state. If the study uses third-trimester BMI, it may overestimate obesity levels and compromise the accuracy of the findings. Also, could pregnancy-related hormones influence the characteristics of adipose-derived stem cells? Can these results be generalized to represent adipose stem cell outcomes in non-pregnant populations across different BMI ranges?

2. The authors characterized ASCs only by flow cytometry. Differentiation assays (osteogenic, adipogenic, chondrogenic) are required to confirm MSC identity per ISCT guidelines.

3. Fig. S2D indicate significant decline in the adipose differentiation rates of obese ASCs, are there alterations in their osteogenic and chondrogenic differentiation functions? And whether this is a universal phenomenon among normal individuals.

4. The causal relationship between ECM remodeling, TGF β activation, and ciliary dysfunction remains unclear.

5. According to Figure 5A, primary cilia appear to localize to the nuclear surface? Could higher-quality images be provided.

6. ADCY3-cAMP activation only validates ciliary length and motility but lacks downstream signaling (e.g., cAMP-PKA) or differentiation capacity (adipogenic, osteogenic and chondrogenic) assessment.

Version 1:

Reviewer comments:

Reviewer #1

(Remarks to the Author)

In this revised version of the manuscript entitled "In-Depth Analysis of Obesity-Associated Changes in Adipose Tissue-Derived Mesenchymal Stromal/Stem Cells and Primary Cilia Function" Kreis and colleagues provide additional data that strengthens their approach. All comments have been addressed accurately with new data or text modifications. Of note, the concerns about statistics and control experiments have been solved. The manuscript is now better organized and reads well. The conclusions drawn by the authors are better aligned with the main data.

I can only be positive and acknowledge the work provided for the revision and the overall improvement of the current manuscript. I think this paper would be a good fit for the journal.

Reviewer #2

(Remarks to the Author)

The authors have adequately addressed all my comments and implemented the suggested revisions. I have no further concerns, and I believe the manuscript is now suitable for publication.

Reviewer #3

(Remarks to the Author)

The authors have addressed the vast majority of the concerns raised in the previous round of review. The revisions have significantly improved the clarity, rigor, and overall quality of the manuscript.

Rebuttal Letter

Dear Reviewers,

We sincerely thank you for your constructive feedback, which has helped us to substantially improve our manuscript. In response, we have performed ten additional experiments (Fig. 5E-G: one experiment; Fig. 6A-G: three experiments; Fig. S2G-H: one experiment; Fig. S3A-H: three experiments; and Fig. R1 (for Reviewer): one experiment). All experiments were independently conducted in triplicate to ensure robustness of our results. Moreover, we have carefully re-analyzed the data using the appropriate statistical tests, reorganized the figures, and thoroughly revised the manuscript text. Detailed responses to each of your valuable comments are provided in the point-by-point rebuttal letter below.

Reviewer #1 (Remarks to the Author):

Summary

In this manuscript entitled “In-Depth Analysis of Obesity-Associated Changes in Adipose Tissue-Derived Mesenchymal Stromal/Stem Cells and Primary Cilia Function”, Kreis et al. describe the transcriptomic profile occurring in primary adipose tissue progenitors of individual with and without obesity. They first put the focus on alterations in the expression of genes involved in the extracellular matrix and adhesion molecule compositions, and their master regulator, the TGF β signalling pathway. In the second part of the manuscript, the authors switch the focus on primary cilia genes and cilia structure and assembly, demonstrating alterations in adipose tissue progenitors during obesity. Of note, the length and mobility of primary cilia are reduced, and the authors propose that alteration in the ADCY3-cAMP signalling cascade mediates these alterations.

Comments:

1) In Figure 1, panels A and C do not provide information regarding the dataset, this should be placed in supplementary. Line 205, the term “relevant” is used, what does “relevant” mean?

We thank the reviewer for this suggestion. In the revised manuscript, the heatmap has been removed. Regarding the term “relevant”, we have replaced it with “significantly deregulated genes ($p < 0.05$)” to enhance clarity and scientific accuracy (lines 246-247).

2) In Figure 1, the authors should better characterise the pathway analysis in correcting for redundancy in the list of genes contained in each pathways, by using PathFinder or an equivalent tool that corrects for such an overlap in terms. For example, in 1D, the authors state that multiple pathways linked to the ECM are identified “including genes responsible for the ECM in general (69 genes), proteinaceous ECM (56 genes), extracellular structure organisation (54 genes), and ECM organisation (47 genes).” It is very likely that the gene list of the different “pathways” overlap and this wrongfully emphasise the data, please show a Venn diagram of the common genes belonging to multiple pathways. Same for 1E, F and G. It is very likely that TGF β related pathways will be identified, as it is well described as a regulator of the ECM and is increased in adipose tissue during obesity, rather than manually isolating the term and depicting a few genes.

We thank the reviewer for this constructive comment. To address this issue and to improve our pathway analysis, we have taken the following steps:

1. Redundancy correction: we utilized the Revigo tool (Supek et al., 2011; PMID: 21789182) to reduce redundancy among GO terms and to summarize enriched biological processes.

2. Visualization of gene overlap: to better illustrate the extent of gene overlap among ECM-related GO terms, we used InteractiVenn (Heberle et al., 2015; PMID: 25994840), as suggested. We have added two Venn diagrams (now presented as Supplementary Figure S1A and B) to clearly show the overlap genes among the selected GO terms.
3. Revision of figures and text: Based on this analysis, we revised the pathway figures and the corresponding text to avoid overemphasizing overlapping GO terms (lines 249-258).
4. TGF β pathway: the TGF β signaling pathway was indeed identified in the GO analysis of the ASCsub40 group and is now included in the revised Fig. 1D (formerly Fig. 1E). For the ASCsub35 group, we have retained a manual depiction of TGF β -related genes to facilitate the comparison with ASCsub40, shown in the revised Fig. 2.

The revised version better reflects the underlying biology, without redundancy and overinterpretation.

3) Figures 2, 3 and 4 are an overuse of transcriptomic data. The tables and volcano plot are fully redundant in the information provided to the reader. The RT-PCR validation should, at best, be provided as supplementary data. Figures 2, 3 and 4 are an extension of a currently incomplete Figure 1, and simply illustrate genes belonging to the pathways identified in Figure 1. Additionally, an increase in ECM production and TGF β signalling in adipocyte progenitors during obesity is already well characterised in both mice and humans (PMIDs : 22355352, 17065356, 18945811, 36617688, 30530781, 16253647, 10923640, 23449528). As a result, the main conclusions from the paragraphs 1 and 2 are not only redundant, but also only focus on known alterations during obesity.

We thank the reviewer for this critical comment. We are aware that increased ECM production and enhanced TGF β signaling in adipocyte progenitors during obesity have been previously reported in both mice and humans, and some of these important references are cited in the revised manuscript (line 287 and 491).

Based on these established findings, our major goal is to identify and compare specific gene sets between ASC subpopulations, which serve as the basis for the later functional analyses of this work.

To avoid redundancy and streamline the figures, we have made the following changes:

1. Figure adjustment: the previous Supplementary Figure 1 has been removed. The former Figure 2, containing RT-PCR validation, has been moved to the supplementary section and is now Supplementary Figure 1C to G (from line 268 onwards).
2. Retention of key figures: we have retained the former Figures 3 and 4 (now Figures 2 and 3), as they illustrate gene expression differences in key pathways related to TGF β signaling, motility and migration. These figures are critical to the mechanistic progression of the manuscript and provide a necessary link between the transcriptomic data and the functional analyses presented later.

4) Figure S2, an alteration of the adipogenic potential of AT progenitors from obese individuals compared to lean is already well described, especially through increased TGF β signaling (PMID : 30610207). Here the authors show quite a strong impairment, which might not reflect the reality of the pathophysiology of obesity. It would be reassuring to see data of differentiation using more conventional differentiation protocols with priming.

According to this comment, we used the standardized StemMACS™ AdipoDiff Medium (Miltenyi Biotec, #130-091-677), which is designed to support efficient adipogenic differentiation of human ASCs. The observed reduction in differentiation was consistent across donors and reflects intrinsic

differences between ASC subpopulations under standardized conditions. We have addressed this point in the protocol of the method section (line 123-124) and provided additional representative images in Supplementary Figure S2.

As described in the above-mentioned literature (PMID: 30610207), reduced adipogenesis in obesity is mediated not only by TGF β signaling alone but also by multiple pro-inflammatory cytokines such as IL-6 and TNF- α . Our earlier work showed that ob-ASCs secrete higher levels of pro-inflammatory cytokines, and exhibit impaired responsiveness to osteogenic differentiation cues due to shortened primary cilia (Ritter et al., 2018, PMID: 29396182).

To further support our findings, and in response to reviewer's comment #7, we repeated the adipogenic differentiation for RT-PCR analysis with ASCs from three donors and replaced the original data (Figure S2F). Our results are consistent with our previous study (Ritter et al., 2018, PMID: 29396182), in which subcutaneous ASCs from obese donors showed significantly reduced adipogenic potential (16.3%) compared to ASCs from lean donors (43.7%). In the present study, we observed a similar pattern: 56.5% differentiation in lean ASCs versus 17.0% in obese ASCs (Supplementary Figure S2D).

5) In Figure 7, what is the rationale between treating each group with only 1 compound? Do the authors claim that progenitors from obese individuals have reduced cAMP signalling, and this affects cilia length? As this is not demonstrated, they should treat both groups with both compounds and assess for cilia length (Lean CTL vs SQ vs NKH, and obese CTL vs NQ vs NKH).

We thank the reviewer for this valuable suggestion. Since ADCY3 expression is already reduced in ASCs from donors with obesity, our initial rationale was to rescue the phenotype in obese ASCs using an ADCY3 activator (NKH), and to induce obese phenotype in lean ASCs by inhibiting ADCY3 (SQ). As recommended, we have now extended our experiments to include both compounds in both groups and assessed ciliary length accordingly. These additional data are included in the revised manuscript as Supplementary Figure S3C and D and described in the text (lines 456-460).

We also performed additional functional studies using siRNA-mediated knockdown of ADCY3. Lean ASCs were transfected with siADCY3 or sicontrol (sicon) for 48 h. Knockdown efficiency was confirmed by RT-PCR. Further analyses show that ADCY3 knockdown reduces both ADCY3 expression and ciliary length. These results are included in the revised manuscript (Fig. 5E to G) (lines 412-419), further supporting the involvement of ADCY3-cAMP signaling in ciliary length maintenance.

6) Same in Figure 8, it would be easier to draw conclusions from the mobility measures if both lean and obese cells were treated with the 2 compounds.

We appreciate this comment. We have performed the suggested experiments and the results are now included as Supplementary Figure S3E-H and described in the text (lines 475-478).

7) The authors do not address whether improving ciliation is beneficial for adipogenesis, and thus the statement "The data suggest that impaired primary cilia in ASCs contribute to defective adipogenic differentiation and cell motility" should not be made. The authors should demonstrate that the compounds they use (SQ and NKH) can effectively influence adipogenesis. Furthermore, a proper demonstration can only be made by combining the treatments with a ADCY3 knock-down.

We thank the reviewer for this important and constructive comment. We have modified the word "contribute" with "may/could contribute" throughout the text to better reflect the correlative nature of our findings. This statement is also supported by previous studies demonstrating that primary cilia are critical for adipogenic differentiation of human ASCs (Forcioli-Conti et al., 2015; PMID: 25637533).

Notably, inhibition of cilium elongation via IFT88 knockdown reduced adipogenic differentiation of MSCs (Dalbay et al., 2015; PMID: 25693948). Moreover, we have previously shown that ASCs from donors with obesity exhibit shortened cilia that are unable to respond dynamically to differentiation stimuli (Ritter et al., 2018; PMID: 29396182). We have now included a clearer explanation and appropriate citations in the revised manuscript (line 371-373).

To further address the reviewer's suggestion, we conducted additional experiments to evaluate the functional effects of modulating ADCY3 activity. Specifically, lean ASCs were pre-treated with the ADCY3 inhibitor SQ, and obese ASCs with the ADCY3 activator NKH, 24 hours prior to induction of adipogenic differentiation (Fig. R1A). After 5 days of differentiation, as expected, obese ASCs displayed significantly less adipogenic genes compared to lean ASCs (Fig. R1B). However, in this short-term treatment setting, we observed only trends, not statistically significant changes, in adipogenic differentiation genes in treated lean (Fig. R1C) as well as obese ASC (Fig. R1D). This suggests that a single 24-hour pre-treatment may be insufficient to induce measurable effects on differentiation under these conditions.

Fig. R1

Figure R1: ASCs from patients with obesity have an impaired ability to differentiate into adipocytes. (A) Subcutaneous lean ASCs (ASCsub lean) pre-treated with the ADCY3 inhibitor SQ and obese ASCs

(ASC sub obese) pre-treated with the ADCY3 activator NKH, were induced to adipogenic differentiation (adipo. diff.) for 5 days. (B-D) The gene expression of *ADIPOQ* (adiponectin), *FABP4* (fatty acid-binding protein 4), *LPL* (lipoprotein lipase), and *PPARG* (peroxisome proliferator-activated receptor gamma) are shown for undifferentiated (con) and differentiated (diff) ASCs (B), lean vs. lean pre-treated with SQ (C), and obese vs. obese pre-treated with NKH (D). The results were obtained from three individual donors in each group and presented as relative quantification (RQ) with SEM. *GAPDH* served as endogenous control. The ordinary one-way ANOVA followed by Tukey's multiple comparisons test was used to assess statistical significance between the groups. * $p < 0.05$, *** $p < 0.001$

8) The authors already published that ciliation is impaired in adipose progenitors from individuals with obesity (Ref 14). They propose that decreased gene expression of the ciliary component shown here is the underlying mechanism. Gene expression alone is not sufficient to draw functional conclusions. Is reduced gene expression alone sufficient to reduce cilia formation? What is the mechanism driving this decrease in gene expression of ciliary components? Did the authors measure the protein and activation level of major transcription factors involved in cilium biogenesis such as the RFX family of FOXJ1?

We thank the reviewer for this comment. We have further investigated potential mechanisms underlying impaired ciliation in ASCs from individuals with obesity. Members of the Regulatory Factor X (RFX) family are among the most well-established transcriptional regulators of primary (immotile) ciliary biogenesis (PMID: 24644260; 2227339; Fig. 6A and B in the revised manuscript, lines 421-428). In contrast, *FOXJ1* primarily regulates motile cilia (PMID: 19011630; 32616903), and is therefore less relevant in the context of ASCs that have primary cilia.

To test whether TGF β might drive reduced expression of ciliary genes, we treated lean ASCsub cells with TGF β for 48 h. This treatment led to a measurable reduction in cilium length (Fig. 6C and D). In parallel, RT-PCR analysis showed that TGF β significantly downregulated expression of *RFX2* as well as *ADCY3*, along with other crucial genes involved in cilium assembly (*IFT88* and *IFT172*) (Fig. 6E and F). To verify treatment efficiency, we also assessed expression of established TGF β target genes (*COL1A1*, *SERPINE1*), all of which were significantly upregulated, validating the activation of the pathway (Fig. 6G). These findings suggest that altered TGF β -RFX2-ADCY3 network may contribute to impaired ASC cilium formation in obesity (lines 429-440, and 524-535).

9) Throughout the manuscript, there is an obvious attempt to justify the choice of genes plotted from the transcriptomic dataset. This leads to an overwhelming number of references and data for the reader to process, which dilute the main message rather than supporting the authors statements and make the manuscript difficult to read. The authors should narrow their writing and refer to relevant literature only when needed.

We thank the reviewer for this suggestion. In response, we have revised the manuscript and reduce the number of explanations and references throughout the result section.

10) Statistics : Comparison of 3 or more groups can not be tested through multiple Student t-test nor Mann-Whitney. This is an abuse of statistical power. Please use ANOVA or Kruskal-Wallis depending on Gaussian distribution, combined with a post-hoc test for Fig2.E, Fig3.F, Fig4.E, Fig5.B, Fig7.E, F, H and I, and Fig8.C, D and E. Fig.6B and D, and Fig.S2E, F, G and H upper panels should be tested with a 2-Way ANOVA combined with post-hoc tests.

We thank the reviewer for this important and constructive comment. We have carefully re-evaluated all statistical analyses across the figures and updated the Method section accordingly, providing full details of the tests applied for each figure (line 223-238). In most cases, comparisons involving three

or more groups have been re-analyzed using one-way ANOVA (with Tukey's, Dunnett's, or Sidak's post hoc tests) or the Kruskal-Wallis test with Dunn's post hoc test, depending on data normality and variance homogeneity.

An exception is Figure 5B and D (previous Fig. 6B and D), where we analyzed the mean fluorescence intensity between lean and obese groups at each defined percentile (10% intervals) of the relative cilium length (0 to 100%). In this case, we conducted groupwise pairwise comparisons (e.g., lean 10% vs. obese 10%, lean 20% vs. obese 20%, etc.). Since the data were not normally distributed, a Kruskal-Wallis test with Dunn's multiple comparisons was applied at each interval. We chose this method because classical 2-way ANOVA with post hoc tests assumes Gaussian distribution, which was not met in this dataset.

Reviewer #2 (Remarks to the Author):

- Brief Summary of the Manuscript

The manuscript reports novel findings regarding obesity and adipogenic differentiation, a topic of global pertinence. The authors explore primary ciliary dysfunction and its potential implications in obesity, particularly in studying adipose tissue-derived mesenchymal stromal/stem cells (ASCs), a relatively underexplored area of research.

The authors obtained ASCs from women undergoing cesarean section, categorizing them into three groups based on body mass index (BMI) for comparison: lean (BMI

The authors conclude that defective ASCs may contribute to the development of obesity and that their restoration could present potential therapeutic targets for obesity-related disorders.

- Overall Impression of the Work

The study addresses a relevant topic in primary cilia dysfunction in ASCs. However, there are some aspects of the study and methodology that require further clarification from the authors.

- Specific Comments

1. There are inconsistencies in the use and definition of abbreviations throughout the manuscript. For example, 'ECM' is defined only in the abstract and not in the main text, whereas 'AT' is defined in the abstract and introduction. Additionally, both 'PPAR γ ' and 'PPARG' are used inconsistently. Review the manuscript for consistent and appropriate use of abbreviations.

We have modified abbreviations throughout the text. PPARG is used for the gene name in our study and PPAR γ refers to the protein in cited reference (Lee, 2018; PMID: 29409985).

2. In Table 2, the values are presented without specifying the corresponding units (e.g., percentage, arbitrary units, or mean fluorescence intensity...). Please clarify and include the appropriate units.

We thank the reviewer for this observation. It is in percentage, which we have revised in Table 2 (line 149).

3. The manuscript does not provide sufficient detail regarding the clinical background of the donors from whom ASCs were isolated. It would be helpful to clarify whether the donors were otherwise healthy or had any conditions potentially associated with inflammation. For example, were glucose levels assessed? As hyperglycemia during pregnancy is relatively common and may influence inflammatory status, such information could help interpret the data more accurately.

We have now provided the clinical background of the donors: all participants were otherwise healthy and showed no evidence of inflammation or impaired glucose tolerance, as confirmed by the 50 g oral glucose tolerance test (oGTT50) and, if indicated, an oGTT75 or hemoglobin A1c (HbA1c), as specified (lines 100-102).

4. It is not clear whether the authors tested the viability of ASCs upon treatment with SQ22536 and NKH477 in their model. In addition, the final concentration of DMSO should be reported, as DMSO can interfere with multiple cellular pathways.

We thank the reviewer for this important comment. We have now assessed the viability of ASCs following the treatments, and the corresponding data have been provided as Figure S3A and B, and described in the text (lines 454-458). To match the highest concentration of SQ22536 used, DMSO was applied at a final concentration of 0.05% (v/v), stated in the revised Method section (lines 115-116).

5. Approximately half of the figures in the manuscript are dedicated to transcriptomic analyses, many of which consist of violin plots. The high number of similar plots makes the figures visually dense. The authors may consider simplifying the presentation - for example, by transferring some of the detailed plots to the Supplementary Material.

As recommended, we have rearranged several figures and incorporated new data into the manuscript. The previous Supplementary Figure 1 has been removed. Former Figure 2 has been moved to the supplementary section and is now presented as Supplementary Figure 1C-G.

We have retained the former Figures 3 and 4 (now Figures 2 and 3), as they illustrate gene expression differences in key pathways related to TGF β signaling, motility, and migration. These figures are critical to the mechanistic flow of the manuscript and provide an essential link between the transcriptomic data and the functional analyses presented in the subsequent sections.

Reviewer #3 (Remarks to the Author):

This study systematically investigates the impact of obesity on the functionality of adipose-derived mesenchymal stromal/stem cells (ASCs), focusing on extracellular matrix (ECM) remodeling, TGF β signaling, ciliary dysfunction, and cell motility. By integrating transcriptomic analysis, functional experiments, and pharmacological interventions, the authors reveal that obesity downregulates ADCY3/CLUAP1, leading to shortened cilia and functional impairment, thereby affecting ASC differentiation and migration. This study is interesting with clear logic, however, several critical issues require clarification to strengthen methodological rigor and mechanistic interpretation.

1. Why was subcutaneous fat from women undergoing cesarean section selected? Is the BMI measured during the third trimester or pre-pregnancy? This distinction is critical because weight gain during pregnancy is a normal physiological process, whereas pre-pregnancy BMI more accurately reflects an individual's basal metabolic state. If the study uses third-trimester BMI, it may overestimate obesity levels and compromise the accuracy of the findings. Also, could pregnancy-related hormones influence the characteristics of adipose-derived stem cells? Can these results be generalized to represent adipose stem cell outcomes in non-pregnant populations across different BMI ranges?

We thank the reviewer for these important comments. Subcutaneous fat samples were obtained from women undergoing cesarean section, as our research is conducted in the Department of Gynecology

and Obstetrics. The BMI used in the study refers to the pre-pregnancy BMI, which is more reflective of the individual's metabolic baseline. We have now clarified this point in Table 1 (line 105).

Regarding the potential influence of pregnancy-related hormonal changes on ASCs, previous studies have shown that the reproductive state has minimal impact on the differentiation capacity of human ASCs (PMID: 19181742). In addition, we have also been working on ASCs from breast adipose tissue from patients with or without obesity and have observed lots of ASC similarities between these two donor groups. Based on the data, we think that the findings can be reasonably generalized to non-pregnant populations across different BMI ranges. We have integrated this information to the manuscript as well (lines 99-100).

2.The authors characterized ASCs only by flow cytometry. Differentiation assays (osteogenic, adipogenic, chondrogenic) are required to confirm MSC identity per ISCT guidelines.

We thank the reviewer for this important point. As recommended, we now provide gene expression data for master transcription factors *RUNX2* (runt-related transcription factor 2) for the osteogenic and *SOX9* (SRY-Box Transcription Factor 9) for chondrogenic differentiation in the revised version (Fig. S2G and H) (lines 387-392).

3.Fig. S2D indicate significant decline in the adipose differentiation rates of obese ASCs, are there alterations in their osteogenic and chondrogenic differentiation functions? And whether this is a universal phenomenon among normal individuals.

We thank the reviewer for this comment. Yes, in addition to the observed decline in adipogenic differentiation (Fig. S2D), previous studies from our group have shown that ASCs from individuals with obesity also exhibit impaired osteogenic and chondrogenic differentiation capacities. Specifically, in a cohort of breast cancer patients, we found that ASCs from obese donors displayed a significantly reduced ability to differentiate into all three mesenchymal lineages (PMID: 36710348). Furthermore, in an earlier study, we showed that ob-ASCs with shortened cilia are inefficient at responding dynamically to extracellular stimuli like osteogenic differentiation (PMID: 29396182), suggesting a mechanistic link. The data suggest a consistent trend: obesity negatively impacts the differentiation potential of ASCs. In contrast, ASCs from lean individuals typically retain full tri-lineage differentiation capacity under standard culture conditions (lines 386-391).

As mentioned above, we now also provide expression data for the osteogenic and chondrogenic master transcription factors *RUNX2* and *SOX9*, respectively, which are also reduced in obese ASCs upon differentiation (Fig. S2G and H) (lines 387-392).

4.The causal relationship between ECM remodeling, TGF β activation, and ciliary dysfunction remains unclear.

The causal relationship between ECM remodeling, TGF β activation, and ciliary dysfunction is indeed not fully understood. ECM remodeling can influence TGF β activation, often by releasing latent TGF β from the matrix (Deng et al, 2024, PMID: 38514615). Primary cilia may play a role in sensing mechanical and chemical signals from the ECM, and ciliary dysfunction could potentially disrupt these pathways, contributing to abnormal TGF β signaling and further ECM alterations. The ciliary membrane is associated with receptors for extracellular matrix (ECM) proteins, and TGF β receptors as well as other signaling components (Christensen et al., 2017; PMID: 27638178). In murine chondrocytes (Kawasaki et al., 2015; PMID: 25828538) and human osteoblasts (Ehnert et al., 2017; PMID: 28271209), TGF β treatment has been reported to reduce primary cilia length. Ciliopathies, genetic disorders caused by defects in the structure or function of primary cilia, are associated with fibrosis, ECM accumulation,

and enhanced expression of TGF β (Seeger-Nukpezah and Golemis, 2012, PMID: 22819513). However, the precise mechanisms linking these processes remain unclear.

During the revision, we performed new experiments to address the role of TGF β in ciliary biogenesis. Interestingly, as shown in Figure 6, TGF β , increased in obesity, is able to shorten the cilium length of lean ASCs. Mechanistically, TGF β efficiently downregulates the gene expression of *RFX2*, a potential transcription factor for *ADCY3*, and *ADCY3*, a key player in ciliary biogenesis, in addition to downregulating other two fundamental ciliary regulators *IFT88* and *IFT172*. These new data highly indicate that the TGF β -RFX-ADCY network is of crucial importance for the maintenance of functional primary cilia in ASCs. We have addressed this point in the Results and Discussion section of the revised manuscript (lines 429-440, and 524-535).

5. According to Figure 5A, primary cilia appear to localize to the nuclear surface? Could higher-quality images be provided.

We thank the reviewer for this comment. We provide images generated by overlaying individual confocal Z-sections using maximum intensity projection from confocal laser scanning microscopy, detailed in the Method section (lines 188-192).

Since primary cilia originate from the basal body (centrosome) as a protrusion of the plasma membrane, their apparent localization can vary depending on the orientation and level of the Z-stack. In some cases, this may give the impression that the cilium is directly localized to the nuclear surface.

6. ADCY3-cAMP activation only validates ciliary length and motility but lacks downstream signaling (e.g., cAMP-PKA) or differentiation capacity (adipogenic, osteogenic and chondrogenic) assessment.

We appreciate this comment. As mentioned above (to comment 7 of Reviewer 1), we conducted additional experiments to evaluate the functional effects of modulating ADCY3 activity. Specifically, lean ASCs were pre-treated with the ADCY3 inhibitor SQ, and obese ASCs with the ADCY3 activator NKH, 24 hours prior to induction of adipogenic differentiation (Fig. R1A). After 5 days of differentiation, as expected, obese ASCs displayed significantly less adipogenic genes compared to lean ASCs (Fig. R1B). However, in this short-term treatment setting, we observed only trends, not statistically significant changes, in adipogenic differentiation genes in treated lean (Fig. R1C) as well as obese ASC (Fig. R1D). This suggests that a single 24-hour pre-treatment may be insufficient to induce measurable effects on differentiation but increase the gene expression for adipogenic differentiation.